**communications** engineering

# 3D vector field-guided toolpathing for 3D bioprinting
Meghan Rochelle Griffin [1,5], Spencer E. Bertram [2,5], Noah P. Robison [1], Angela Panoskaltsis-Mortari [3], Ravi Janardan [4] & Michael C. McAlpine [2] ✉

Complex fibrous microarchitectures are common in biology, with fiber orientation playing a key role in the structure–function relationships that govern tissue behavior. Directional imaging modalities, such as diffusion tensor magnetic resonance imaging (DTMRI), can be used to derive a 3D vector map of fiber orientation. Incorporating this alignment information into engineered tissues remains a challenging and evolving area of research, with direct incorporation of directional imaging data into engineered tissue structures yet to be achieved. Here we describe an algorithmic framework, entitled Nonplanar, Architecture-Aligned Toolpathing for In Vitro 3D bioprinting (NAATIV3), which processes DTMRI data to map tissue fibers, reduce them to a representative subset, remove conflicting fibers, select a printable sequence, and output a G-code file. DTMRI data from a human left ventricle was used to 3D print fibered models with high accuracy. It is anticipated that NAATIV3 is generalizable beyond the cardiac application demonstrated here. Directional imaging data from a variety of organs, disease states, and developmental timepoints may be processable by NAATIV3, enabling the creation of models for understanding development, physiology, and pathophysiology. Furthermore, the NAATIV3 framework could be extended to bioengineered food manufacturing, plant engineering, and beyond.

Structure–function relationships are central to all biological systems. Many tissues possess a unique fiber alignment or cellular directionality that is essential to their performance[1–3]. Recapitulating this microarchitecture in vitro is key to generating functional engineered tissues, and this remains a challenging and evolving area of research. Most methods of engineering cellular alignment, including photolithography, directional freezing, acoustic or magnetic patterning, electrospinning, and mechanical loading, produce a uniform, planar alignment that fails to recapitulate the gradations in fiber orientation seen in vivo[4–10]. Additionally, much of this work has been confined to monolayer cultures[1,10]. A need exists for techniques that produce alignment in three-dimensional tissues, which are known to outperform two-dimensional cell cultures in physiologic accuracy and function[11–13].

Additive manufacturing techniques are becoming increasingly popular in engineering 3D organ and tissue models[14–16]. Extrusion-based 3D bioprinting in particular allows for precise spatial control over material deposition that enables the replication of complex architectures without the inherent planar constraints present in many other fabrication processes. This lends itself to producing exogenic organs for physiological modeling, therapeutic testing, and potentially transplantation. A particular advantage of extrusion-based 3D bioprinting is that cells embedded within a bioink are shear-aligned in the direction of material deposition during the extrusion process, prompting a fiber-like organization[5,9,17].

Historically, 3D printing has consisted of uniform material deposition within successive, planar slices of a three-dimensional object. Consequently, when this method is applied to 3D extrusion bioprinting, cell alignment is also limited to these planes, creating a series of stacked monolayers that fail to recreate the native 3D fiber architecture[14]. In order to capture the full complexity of fibrous tissue, material deposition in 3D bioprinting must be extended beyond traditional planar constraints. Specifically, a method is needed to translate information about native fiber structure into commands executable by a 3D printer.

Diffusion tensor magnetic resonance imaging (DTMRI) is a clinically relevant imaging modality that measures the diffusion of water in 3D space[18–21]. Because cell membranes are polar, water preferentially diffuses along cells rather than across the membrane. In tissues where elongated cells arrange end-to-end to form fibers, the diffusion tensor has directionality, which allows for the construction of a vector field representing the fiber direction[22]. Despite the availability of these data, a

[1]Department of Biomedical Engineering, University of Minnesota, Minneapolis, MN, USA. [2]Department of Mechanical Engineering, University of Minnesota, Minneapolis, MN, USA. [3]Department of Pediatrics, University of Minnesota, Minneapolis, MN, USA. [4]Department of Computer Science & Engineering, University of Minnesota, Minneapolis, MN, USA. [5]These authors contributed equally: Meghan Rochelle Griffin, Spencer E. Bertram. ✉e-mail: mcalpine@umn.edu

satisfactory methodology for generating 3D printing toolpaths guided by an input three-dimensional orientation field has not yet been established. Previously proposed methods have been constrained to planar input fields, rendering them insufficient for biological applications[23,24].

We have developed an algorithm, NAATIV3 (pronounced "native"), for generating a sequence of non-interfering, 3D-printable toolpaths from an input vector field. NAATIV3 first numerically integrates the vector field within a boundary volume, producing a dense set of 3D contours, called streamlines, which represent the native fiber structure. The complete streamline set is then reduced to a density capable of fabrication within the resolution limits of the 3D bioprinter. Any potential interference between the physical printing system (i.e., the needle used for extrusion) and previously deposited material during the printing process is eliminated, further reducing the toolpath set to a final subset. Lastly, a greedy search algorithm is used to order toolpaths into a printable sequence, which is then converted into a file format accepted by the 3D printer (i.e., G-code).

The capabilities of NAATIV3 were demonstrated by producing G-code for fabricating a left ventricle (LV) model based on DTMRI imaging of an ex vivo human heart. The LV is of particular interest as a cardiac tissue model because it is the thickest, strongest chamber of the heart and is implicated in many disease states[25]. The efficiency of the LV is related to its cardiomyocyte (CM) fiber structure, which is characterized by a helical alignment that shifts transmurally[21,26,27]. This alignment is essential to the force generation, anisotropic electrical conduction, and biomechanical properties of the myocardium, which collectively contribute to a healthy ejection fraction, a measurement of the amount of blood pumped with each

contraction of the ventricle[26,27]. Currently, engineered ventricle models do not successfully incorporate native CM alignment, contributing to low contractile force and inadequate (<60%) ejection fraction[8,14,28].

For the first time, using NAATIV3, we have demonstrated that imaging data can be computationally manipulated to enable the creation of hyper-realistic models of fibered tissue via extrusion 3D bioprinting. This process transcends previous reliance on planar 3D printing, eliminates unidirectional alignment constraints, and bypasses manual assembly methods in favor of an automated process. Precisely aligned engineered myocardium is the key to exogenic hearts, which have the potential to save thousands of lives per year through medical intervention research and, potentially, transplantation. The development of NAATIV3 represents a major advancement in the biomanufacturing process, moving the field of tissue engineering closer to producing engineered products that accurately recapitulate the structures and functions of native tissues.

## Results
### NAATIV3 algorithm
The NAATIV3 framework was developed to fill the technological gap between directional imaging modalities and additive manufacturing of fibered constructs. Most toolpathing processes for additive manufacturing are generative, meaning that toolpaths are added to an empty volume of delineated space until a target density is reached (Fig. 1a). NAATIV3, instead, uses a reductive methodology, meaning that it begins with a highly dense set of potential toolpaths derived from input data—such as tissue fiber orientation maps—and systematically reduces them to a sequence of 3D-

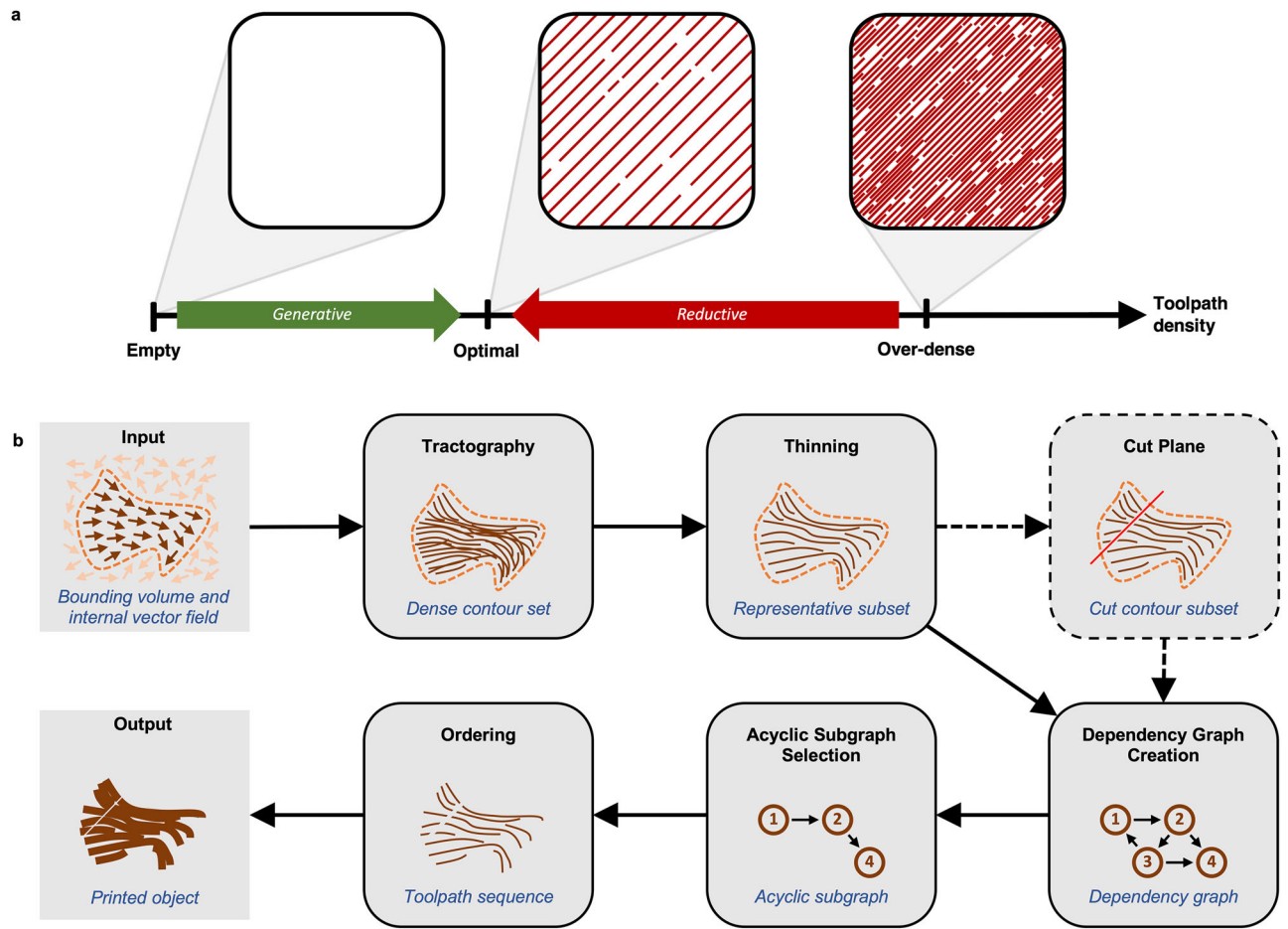

**Fig. 1 | NAATIV3 algorithm overview. a** Alternative toolpathing approaches to achieve optimal toolpath density. Traditional toolpath planning techniques are generative, whereas NAATIV3 employs a reductive approach. **b** Overview of the algorithm used to convert raw directional data to G-code suitable for use in an extrusion-based 3D bioprinter. Schematic input data and output toolpaths shown. Schematic contour sets (brown) inside an arbitrary bounding volume (orange dashed) and directed dependency graphs shown between intermediate steps. Note that the cut plane step is optional, at user discretion.

printable toolpaths (Fig. 1b). In direct contrast to traditional 3D printing, NAATIV3 does not use planar slicing, instead determining toolpath trajectories solely from an input vector field. The directional data is voxelized, where each voxel is a uniform volume that contains a subset of the data. Data processing is performed within a user-specified region of interest, delineated using a 3D voxel mask defined by anatomical markers in the tissue scan (Supplementary Fig. 1A, B).

NAATIV3 employs multiple novel processes to effectively translate tissue fiber data into G-code. One such process allows a highly dense, native fiber set to be reduced to a uniformly spaced subset that is optimized to the parameters of a specific 3D bioprinting system. Another predicts and eliminates interference between toolpaths to prevent the 3D bioprinting equipment from damaging the construct as it is manufactured. A custom heuristic function for the greedy search procedure used to order toolpaths was developed to reduce the total print time by minimizing non-extruding travel distance, while simultaneously guaranteeing an interference-free print sequence, as predicted by a geometric interference model[29].

Each step of the NAATIV3 process is described in detail in the following sections. Further details regarding the mathematical and computational implementation of NAATIV3 can be found in the Supplementary Methods.

## Tractography

With the end goal of creating toolpaths for a 3D bioprinter, directional imaging data must first be processed to create a map of native fiber architecture, a method termed tractography. NAATIV3 performs tractography on discrete vector field data within a region of interest specified by a user-input bounding volume (Supplementary Fig. 1C). Uniformly spaced seed points are generated throughout the volume. Beginning at each seed, fourth-order Runge-Kutta path integration is performed along the input vector field, yielding a dense set of streamlines representing the native fiber structure of the imaged tissue.

To illustrate this, the fiber direction field was extracted from DTMRI imaging of a fixed, ex vivo LV from a human without a history of cardiovascular disease (Supplementary Fig. 1D). One set of data was processed at 1:1 scale, and a second set was resampled to be at 1:4 scale (a practical size for an in vitro biological model). Data was input to NAATIV3 and used to perform tractography, resulting in a set of streamlines representing the LV's fiber structure (Fig. 2a). Fiber architecture in the myocardium is commonly characterized by helical angle (HA), defined as the angle of inclination between a myofiber and the short-axis plane (Supplementary Fig. 1E)[30]. A healthy transmural HA range from endocardium to epicardium is +60° to −60°[21]. The generated streamline set displayed the anticipated smooth transmural progression of helical angle, attesting to the success of the NAATIV3 tractography process (Fig. 2b).

DTMRI data from a heart with a history of myocardial infarction were also run through the NAATIV3 tractography procedure to show the contrasting turbulent and disorganized fiber structure caused by the disease state (Supplementary Fig. 2)[31,32]. This substantiates NAATIV3's ability to perform tractography on arbitrary 3D vector fields from a variety of physiological states. Relevant medical histories for both subjects are provided in Table S1. Hereafter, only the use of NAATIV3 with the healthy LV myocardium is highlighted.

## Streamline set thinning

The resolution of extrusion-based 3D printing is limited by the diameter of the needle used, commonly in the range of 0.1–1.5 mm, whereas tractography has no such physical constraints. Consequently, a reduction of streamline density is required to successfully produce toolpaths for fabrication. The novel sweep exclusion method employed by NAATIV3 reduces the full streamline set to a thinned subset, targeting an optimal packing density based on the resolution of the printer. The density is determined by an inter-streamline spacing parameter, $w_s$, set by the user. For optimal streamline packing, $w_s$ should be equal to the diameter of the printed material. In the ideal case, where the input streamline set is locally parallel,

the resulting subset would have an inter-toolpath distance equal to $w_s$. In real tissue, fibers are not perfectly parallel, in which case $w_s$ becomes the minimum spacing between toolpaths.

Beginning with the longest streamline, a bounding cylinder approximating the geometry of the physical print line is created by sweeping a circular cross-section with radius $w_s$ along the streamline of interest (Fig. 2c). Any streamline that intersects with this cylinder is removed from the set (Fig. 2d). In effect, this serves to average the trajectories of several tightly packed streamlines into a single, representative streamline. After the first streamline is swept, sweep exclusion proceeds to the nearest-remaining streamline. By beginning with the longest streamline in the set, longer streamlines are preferentially selected next, which increases the average toolpath length in the final set, yielding a more contiguous printed structure. The process of creating bounding cylinders, removing intersecting streamlines, and selecting the next streamline to sweep is repeated until all remaining streamlines in the set have been swept or removed (Fig. 2e, Supplementary Video 1).

Tractography of the 1:4 scale LV data produced 425,891 streamlines with 1 seed per $0.2 \times 0.2 \times 0.2$ mm voxel. Tractography of the 1:1 scale model produced 27,220,566 streamlines with 64 seeds per $0.8 \times 0.8 \times 0.8$ mm voxel. Seed densities were selected to yield identical spatial seed densities of 125 seeds per mm². The extrusion needle used for printing was 22 gauge (GA) (0.41 mm inner diameter (ID)) for the 1:4 scale model and 18 GA (0.84 mm ID) for the 1:1 scale model. To determine $w_s$, a sample of the material was 3D bioprinted and imaged with micro-computed tomography (micro-CT). The actual diameter of the extruded material was 0.54 mm with the 22 GA nozzle and 1.0 mm with the 18 GA nozzle, which became the respective value of $w_s$ for each model. The sweep exclusion process yielded thinned sets ($n = 382$ for 1:4 scale, $n = 1,896$ for 1:1 scale), which provide the initial toolpath set for further processing steps. In both sets, a marked decrease in streamline density from the initial to the thinned set is apparent, while the overall ventricle shape and fiber pattern are preserved. These results show that although the thinning procedure reduces the number of fibers included in the final toolpath set, the structural validity of the set is maintained (Supplementary Fig. 3).

## Toolpath interference prediction

To directly use the thinned streamline set as toolpaths for 3D extrusion printing, an ordered sequence must exist such that, during material deposition along a specific toolpath, the extrusion needle does not intersect with material deposited along any of the previously printed toolpaths. NAATIV3 utilizes a predictive model, which relies on a dependency graph, to define relative ordering restrictions (directed edges) between each pair of toolpaths (vertices)[33]. A directed edge between two toolpaths indicates that the first toolpath must be printed after the second toolpath in order to avoid interference during material deposition (i.e., the first toolpath "depends on" the second).

To create the dependency graph, a geometric model for the extrusion needle and the deposited print line material is required to model interference during the 3D printing process. For the extrusion setup used in this study, a vertical cylinder represents the needle, and the deposited ink is taken to have an elliptical cross-section determined by the cutting edge of the needle (Fig. 3a, b). For each directed pair of toolpaths, a volume representing the space spanned by the needle while printing along one toolpath and another volume representing the deposited material along the other toolpath are constructed. The geometry of these volumes is defined by the needle diameter, $d_n$, and the printed material diameter, $d_p$. If intersection between the two volumes is detected, a directed edge is added to the dependency graph. If no intersection is detected, there is no dependency between the toolpaths and no edge is added. This intersection test is performed for every directed pair of toolpaths to populate the dependency graph.

For the LV models, the parameter $d_p$ was set to reflect the actual width of the extruded material, measured above. The parameter $d_n$ was set to the inner diameter of the needle, rather than the outer diameter, a simplification which resulted in fewer calculated dependencies and an increased number

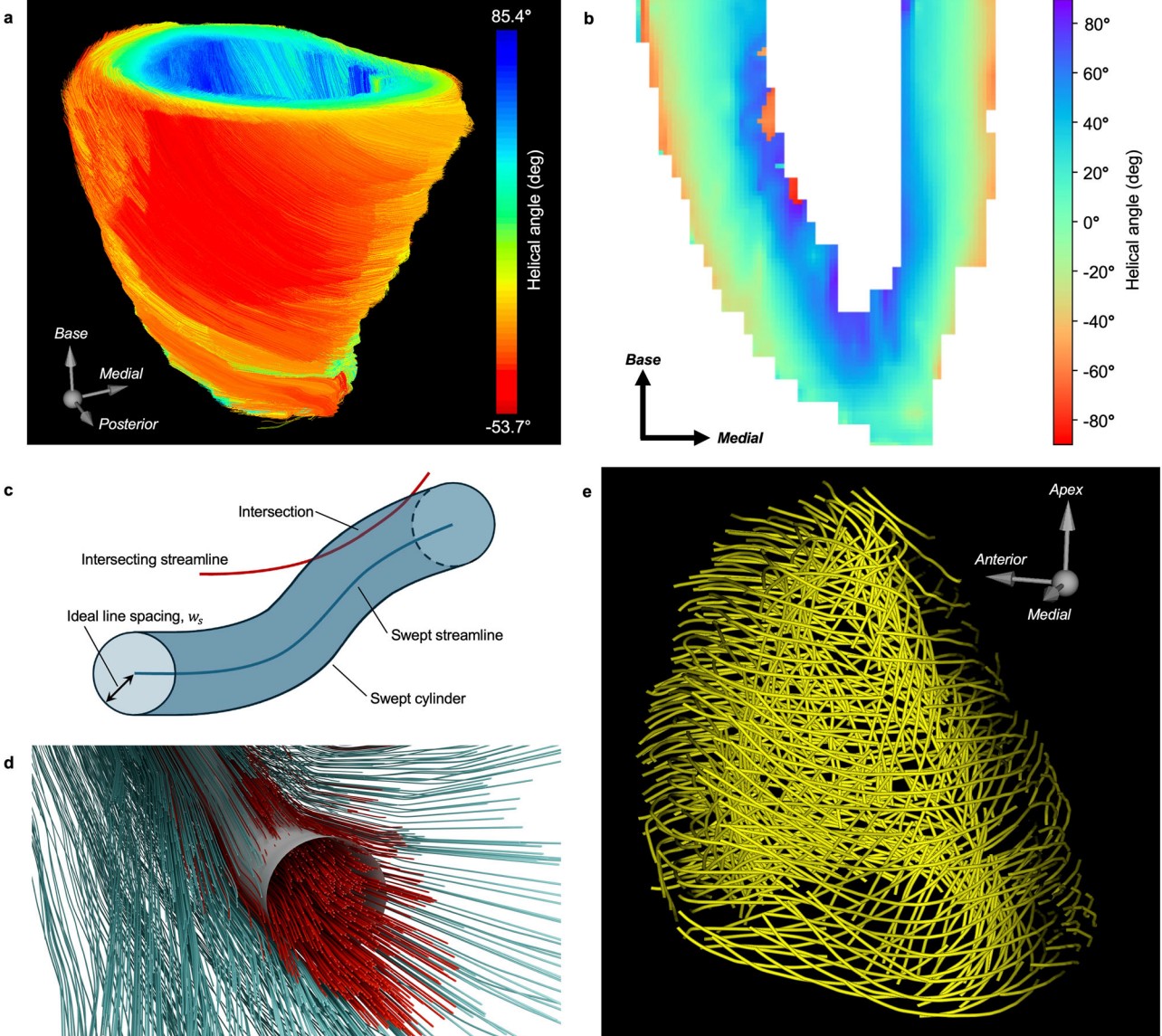

**Fig. 2 | A tractogram of the left ventricle and its representative contour subset.** **a** Complete tractography results for the isolated left ventricle of a healthy human heart, with the free wall visible. Streamlines colored by mean helical angle. Note that streamlines with helical angle above +70° near the endocardium stem from the presence of trabeculations. **b** Vertical cross section of the left ventricle through the septum, depicting the helical angle transition from epicardium to endocardium.

**c** Schematic cylinder intersection during sweep exclusion. The intersecting streamline (red) lies within the swept cylinder (blue) created around the swept streamline (blue) and would be removed during the sweep exclusion process. **d** Rendering of a sweep exclusion iteration, taken from tractogram data. Swept cylinder (gray), removed streamlines (red) and kept streamlines (teal) shown. **e** Representative contour subset yielded by sweep exclusion.

of toolpaths in the final model without substantially impacting the construction of the physical model.

## Resolution of toolpath interference

For the thinned toolpath set to be printable without material interference, the dependency graph must be acyclic. In graph theory, a cycle is a sequence of vertices that starts and ends at the same vertex, in which successive vertices are connected by an edge and there are no repeated vertices. If such a cycle exists in the dependency graph, it is impossible to order the streamlines for printing without inducing interference. NAATIV3 applies a two-part process to ensure that the dependency graph is acyclic.

Depending on the nature of the input vector field, inherent cycles may be present. For example, in cardiac tissue, the helical fiber structure of the LV inherently introduces a large number of cycles[8]. To address this, NAATIV3 includes an optional cut plane step. A plane can be defined that bisects a subset of the toolpaths, effectively splitting those that cross the plane into

two separate toolpaths. The position of the plane can be adjusted to fit the specific fiber structure and geometry of the target tissue. The plane is implemented with zero thickness to retain toolpath density and promote adhesion at the resultant seam. The plane can also be angled to increase the overlapping surface area of the cut ends, further promoting adhesion at the seam. Edges in the dependency graph are explicitly prohibited between toolpaths on opposite sides of the plane. In practice, the cut plane step is implemented prior to the construction of the dependency graph. The need for a manually-defined cut plane geometry is at the discretion of the user and represents a limitation of the NAATIV3 framework.

For the LV models, an angled cut plane positioned in the inter-ventricular septum was implemented (Fig. 3c). The location was chosen to split as few toolpaths as possible, resulting in greater structural integrity in the 3D-printed model. It is expected that the optimal plane location varies between hearts, due to many factors including anatomical differences and age- and disease-related remodeling. To reduce subjectivity in placing the

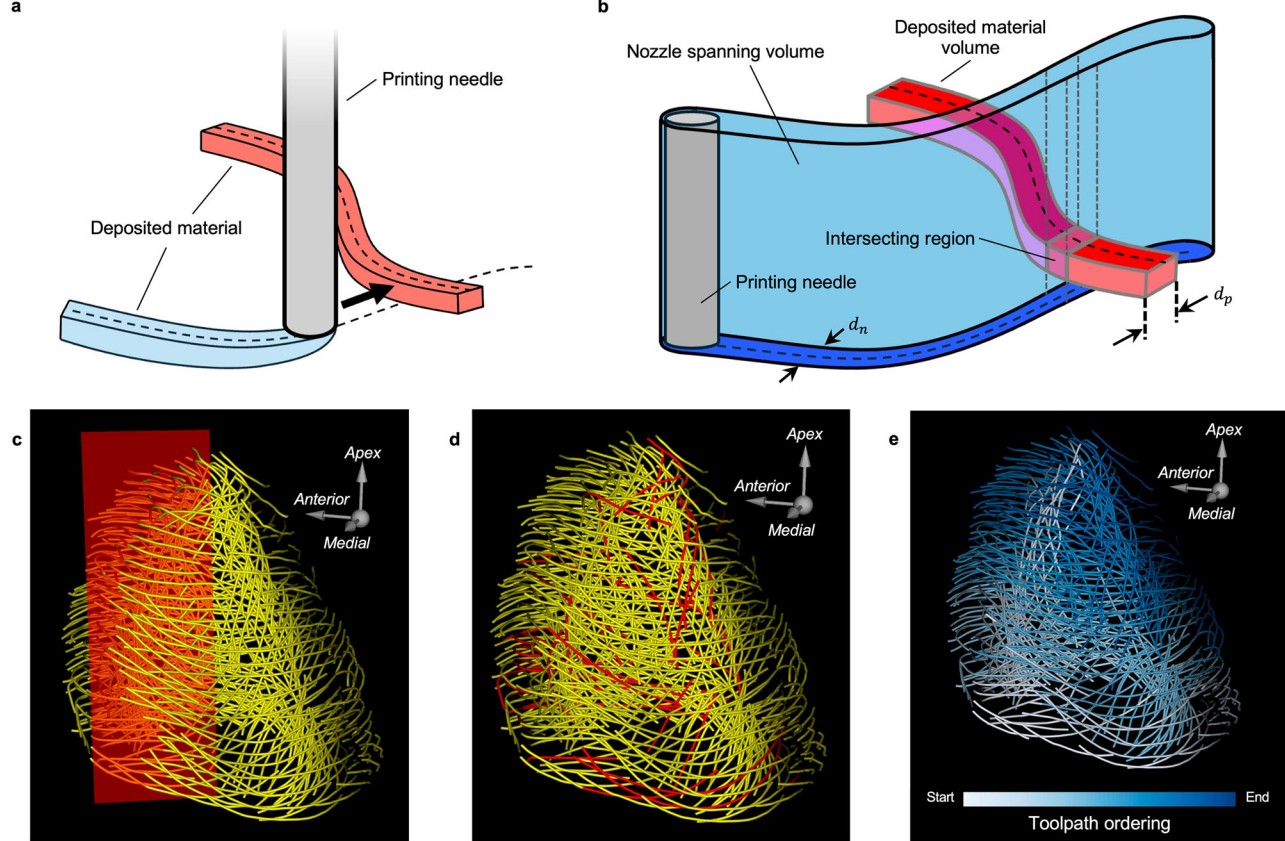

**Fig. 3 | Interfering toolpaths are eliminated to produce a 3D-bioprintable set.** **a** Illustration of the path of an extrusion needle (gray) depositing material (blue). Previously deposited material (red) creates interference. **b** Schematic of volume intersection method used to determine interference between material (red) and the path of the needle (blue). **c** Visualization of cut plane (red) intersecting the thinned toolpath set (yellow) at the interventricular septum of a 1:4 scale human left ventricle model. **d** Acyclic, thinned streamline set (yellow) shown after maximum acyclic subgraph selection, with removed toolpaths in red. **e**) Final toolpath set, colored by greedy search-determined print order.

plane, particularly when processing multiple data sets, quantifiable metrics could be defined to determine the optimal location. These may be the fewest number of toolpaths bisected, fiber orientation patterns, or other application-driven metrics. In non-cardiac applications, where the native fiber architecture may not be predisposed to a large number of cycles, splitting of toolpaths with a cut plane may not be necessary.

Regardless of whether a cut plane is implemented, NAATIV3 employs a pre-processing step and an iterative stochastic optimization step to find the largest acyclic subgraph, yielding a final, non-interfering toolpath set. While the largest acyclic subgraph could theoretically be found exactly, implementing such a method is extremely computationally intensive and, therefore, impractical[34]. In the pre-processing step, bidirectional edges (toolpaths that depend on each other) are identified (Supplementary Fig. 4A, B). For each bidirectional edge, the toolpath deemed more likely to be part of additional cycles, as determined from the dependency graph, is removed from the set. In the subsequent stochastic optimization step, random subgraphs of the full dependency graph are iteratively generated to find the maximum remaining acyclic subgraph. At each iteration, toolpaths are removed at random until the reduced graph is acyclic. This process is repeated for $N$ iterations, and the subgraph with the most toolpaths is retained.

The pre-processing step removed 16 toolpaths in the 1:4 scale LV model and 110 toolpaths in the 1:1 scale model. Diminishing increases in acyclic subset size after the stochastic optimization step were found after 100,000 iterations in the 1:4 scale model and 100,000,000 iterations in the 1:1 scale model (Supplementary Fig. 4C). The acyclic subgraph retained 91.01% of the thinned toolpaths in the 1:4 scale model and 54.9% in the 1:1 scale model. As shown in Fig. 3d, removed toolpaths were evenly

distributed throughout the model and represented a small fraction of overall toolpaths, particularly in the 1:4 scale model. This ensures that the necessary interference removal steps, which ensure printability of the physical model, incur minimal impact on the integrity of the final construct (Fig. 3d).

## Toolpath ordering

To generate the final, printable toolpath print sequence, NAATIV3 uses a greedy search algorithm. Printable is defined as a sequence in which each ordered toolpath does not depend on any toolpaths coming later in the sequence, with the prior interference prediction and removal steps guaranteeing that a printable sequence of toolpaths exists. The heuristic function employed during the greedy search attempts to minimize the non-extruding travel distance, reducing overall print time, which is important for maintaining ink rheology and cell viability for in vitro applications. The coordinates of the ordered toolpath sequence (Fig. 3e) can be written to a text file in the format of G-code. Header and footer commands specific to the 3D printing equipment are added to the file at this time. As guaranteed by the dependency graph and associated cycle-removal procedures, every toolpath in the final sets for the 1:4 and 1:1 scale models of the LV was able to be ordered into a printable sequence.

## 3D printing a fibered ventricle model

To confirm that the G-code generated by NAATIV3 can be successfully manufactured, the ordered toolpaths for the 1:4 and 1:1 scale models were printed using a silicone-based ink, though we anticipate that any liquid bioink could be substituted (Fig. 4a, b, Supplementary Video 2). Because of the complex, non-planar geometries of the toolpaths, the deposited material

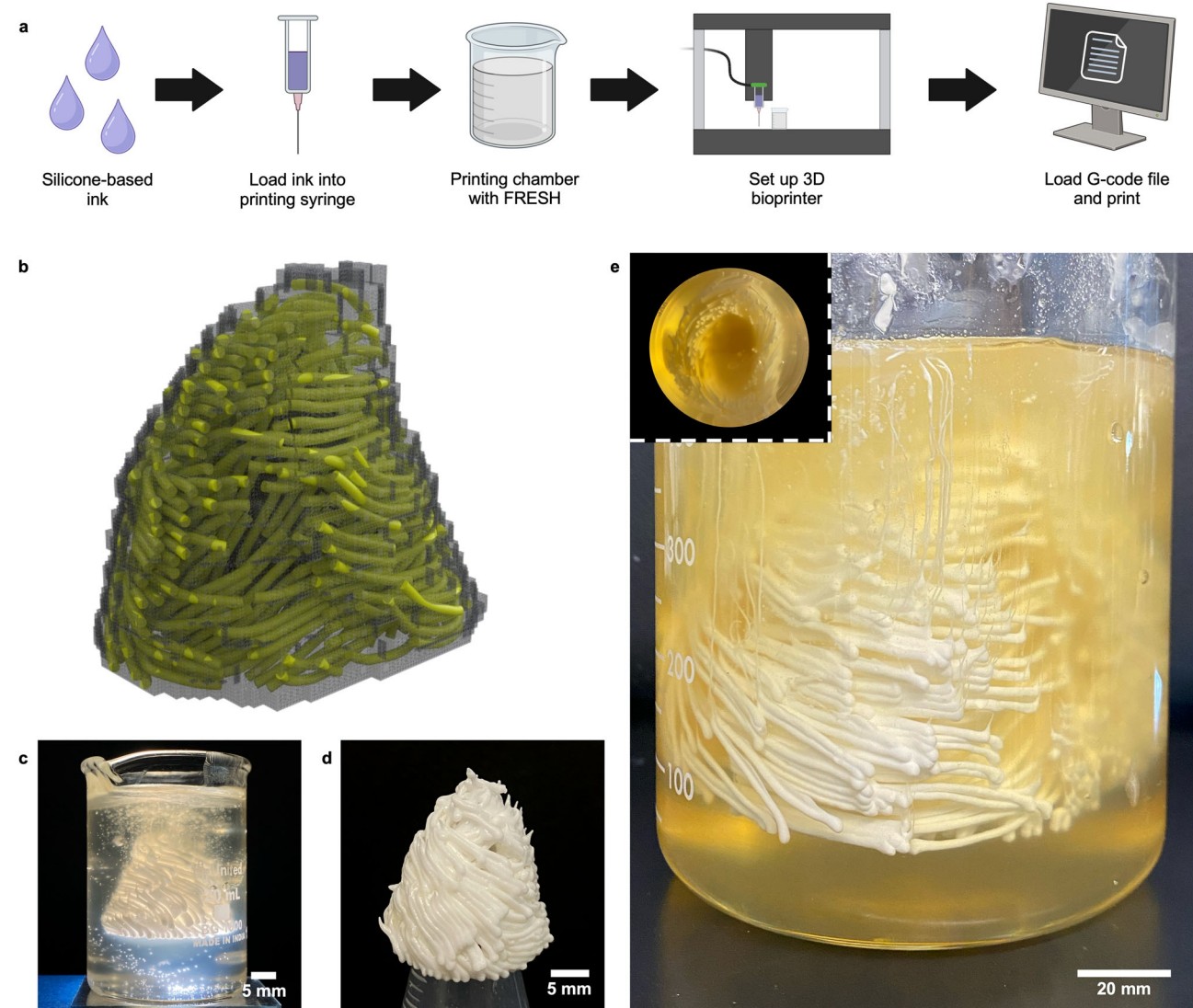

**Fig. 4 | The G-code output by NAATIV3 is 3D bioprinted to produce 1:4 and 1:1 scale left ventricle models. a** Process overview for fabricating a 3D bioprinted, fiber-oriented left ventricle model. G-code is incorporated at the 3D bioprinting step. Created with Biorender. **b** Rendering of the voxel mask (gray) overlaid with the final toolpath set (yellow). **c** The printed structure of the 1:4 scale ventricle model in the support material. **d** The printed 1:4 scale model after extraction from the support medium. **e** The 1:1 scale model in the support material. The inset shows the bottom-up view of the 3D printed model, displaying the cross-section of the ventricle wall at the opening.

was supported during printing using the freeform reversible embedding of suspended hydrogels (FRESH) method[35,36]. The completed 1:4 scale print is shown before (Fig. 4c, Supplementary Video 3) and after (Fig. 4d) extraction from the FRESH support material, confirming printability of the toolpath set and demonstrating the integrity of the final construct. The cured and dried model was easy to handle and durable under manual manipulation. When held upright and filled with water, it appeared watertight. This scale is on par with other LV tissue models that have been reported and is a practical size for in vitro research[8,14,15].

Creating exogenous organs for surgical planning or transplantation requires manufacturing tissues at human scale. Excitingly, the use of NAATIV3 was extended to a full-size LV model, which was successfully 3D printed as a proof of concept for this process at scale (Fig. 4e, Supplementary Video 4). Viewed in the optically clear FRESH support, the overall geometry of the 3D printed, fibered, 1:1 scale model was representative of the source G-code and proportionate to the human heart from which the DTMRI data were obtained. Once removed from the FRESH support, however, the model was unstable and did not hold up to handling.

## Analysis of 3D printing accuracy

The fidelity of the printed construct to the G-code was assessed using a micro-CT scan of the 1:4 scale ventricle model (Supplementary Fig. 5A). Following scan acquisition, a mask isolating the deposited silicone from the support material was defined and exported as an STL file for analysis (Supplementary Fig. 5B-C). The distance between the micro-CT and G-code surface meshes was computed at every point ($n = 1,671,137$) (Fig. 5a). The offset between the models was 0.138 ± 0.250 mm (mean ± standard deviation). Comparatively, the width of the extruded material was 0.54 mm, meaning that the average deviation was less than a quarter of the width of a printed line. The minimum and maximum distances between the models were −0.831 mm and 3.900 mm, respectively. Visual rendering of the results demonstrates a high degree of consistency throughout the printed structure. The primary source of error occurred at the apex of the ventricle model, where shorter streamlines are more vulnerable to displacement after printing, as the FRESH support bath decreases in viscosity at room temperature and the structure is jostled in transport to the micro-CT facility (Fig. 5b). Further segmentation of the micro-CT scan demonstrated the presence of distinct subepicardial, midmyocardial, and subendocardial

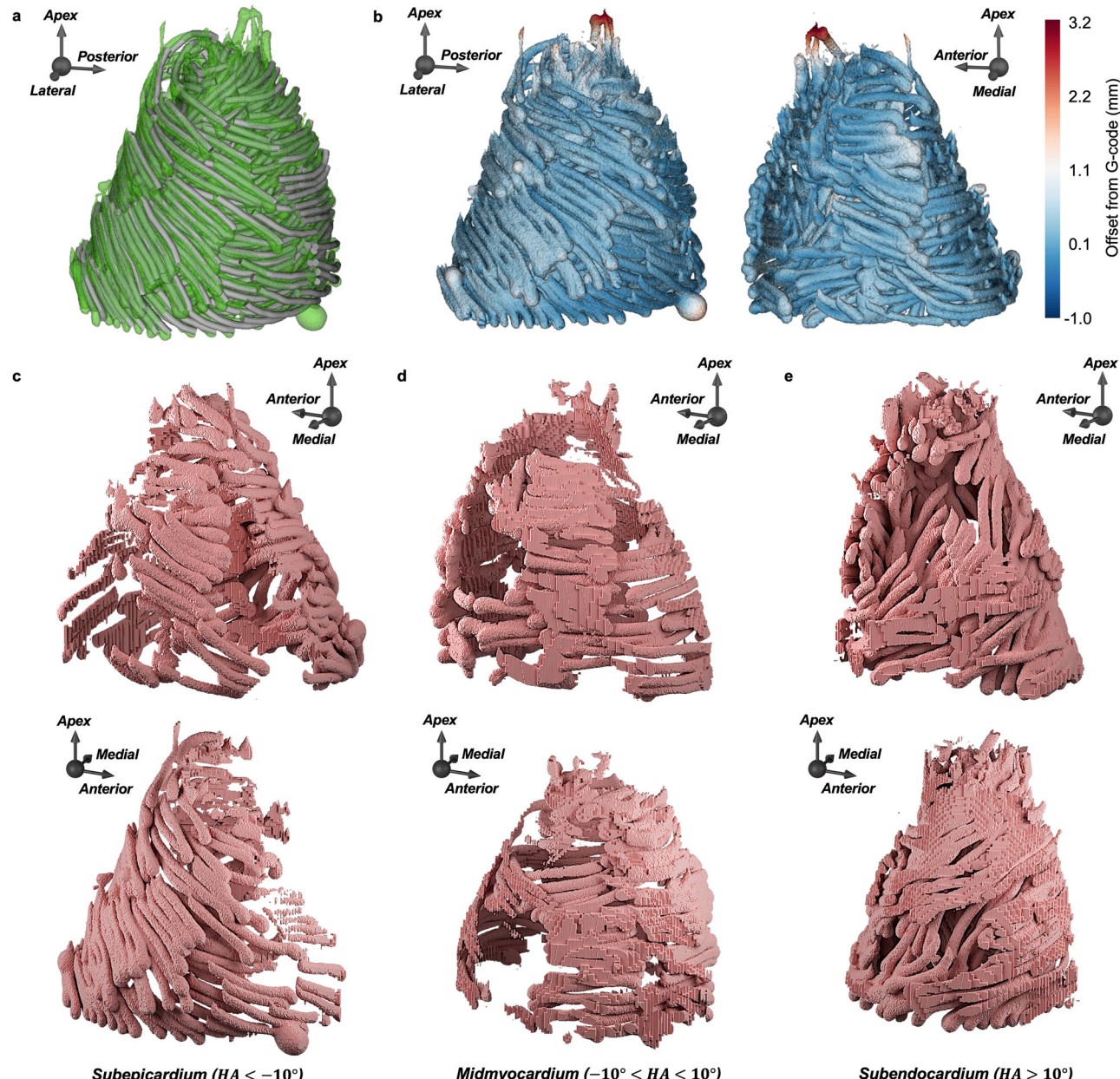

**Fig. 5 | 3D bioprinted constructs are highly accurate to the source G-code. a** STL of the G-code toolpath sequence (gray) overlaid with the STL of the 3D-printed model's micro-CT scan mask (green) for a 1:4 scale human left ventricle model. **b** Offset between G-code and micro-CT. Free wall (left) and interventricular septum (right) views are shown. **c–e** Isolated myocardial regions from the 3D-printed model micro-CT showing the: (**c**) subepicardium; (**d**) midmyocardium; (**e**) subendocardium. Regions were isolated via a combination of filtering based on per-voxel helical angle of the original DTMRI image and manual labeling. Free wall (bottom) and interventricular septum (top) views are shown.

fiber tracts in the final model (Fig. 5c–e). Overall, these results verify that the conversion of G-code into a physical model using extrusion-based 3D bioprinting can be carried out with a high degree of accuracy.

## Discussion

This work introduces NAATIV3, the first method for leveraging directional imaging to create 3D tissues and tissue models that accurately recapitulate native fiber architectures. NAATIV3 begins with 3D vector field data from a target tissue, calculates the fiber tracts, selects a representative subset thereof, detects and removes potential interference based on the 3D printing process, orders the tracts for printing, and produces an output file to be loaded into a 3D printer. NAATIV3 relies on a series of input parameters that can be fine-tuned for specific datasets and printing configurations. Once initialized, the

algorithm runs with little human intervention. As previous methods of encoding fiber alignment into manufactured tissue are limited to planar orientations, this work represents an important step forward in the field of tissue engineering[4–7]. By transcending the planar paradigm and using native fiber tracts to inform toolpath generation, it is possible to 3D bioprint tissue models with a native-inspired fiber structures, which hold potential for in vitro research, preclinical modeling, and surgical planning.

To demonstrate its effectiveness, NAATIV3 was used to generate G-code for the 3D bioprinting of a human LV model, which was chosen because of the helical complexity of its fiber structure, the prevalence of cardiac disease, and the high demand for new therapies. Due to the inability to realistically capture mesoscale patterns of native fiber structure, previously published 3D-bioprinted ventricle models have been unable to

accurately replicate cardiac function, making them poor model systems and insufficient for in vivo implantation[8,14,28]. This work demonstrated the ability of NAATIV3 ability to fabricate organ models in a range of sizes by producing a 1:4 scale model that lends itself to in vitro research as well as a 1:1 scale model that would be required for surgical training or implantation. By replacing the silicone-based ink used here with a cellularized bioink, NAATIV3 could allow for the production of living tissue models, bringing the fields of tissue engineering and 3D bioprinting one step closer to their ultimate goal of fabricating full-scale organs that recapitulate the structure and function of native tissues.

A limitation of the reported work is the lack of cellularization in the 3D printed models. While the NAATIV3 framework and fabrication process are directly applicable to studies incorporating cellular, hydrogel-based bioinks, with appropriate changes made to the printing protocol for the different materials, this was not demonstrated in this paper. Previous reports have demonstrated CM alignment in the direction of material deposition during the extrusion process, and future experiments should be conducted to confirm that this will occur in the fibered LV model as well[37]. Other studies have demonstrated similar alignment results using pre-fabricated microtissues, rather than individual cells, in their bioink, which combine well with the NAATIV3 framework[38]. For the 1:4 scale model, the wall thickness and printing time are not unreasonable for use in cellular experiments and maintenance in cell culture. However, for the 1:1 scale model reported here, both wall thickness and total printing time pose challenges for maintaining cell viability during and after printing.

Variations on the silicone-based, acellular ink used here would be a highly interesting area of future study. Ink modifications could create a wide range of mechanical properties, fully mimicking the mechanical properties of the myocardium at different stages of development and disease. Ink design could also capture the strain-stiffening effects of collagen in the native myocardium. The vast potential for utilizing ink variations with the fibered LV model would allow the creation of highly realistic myocardial models, which are particularly well suited to physician training, surgical planning for personalized medicine, and patient education.

Structure beyond the fiber scale should also be taken into account in discussing the biological and anatomical accuracy of the LV model described here. To accommodate the physical resolution limits of the 3D bioprinter, NAATIV3 includes a sweep exclusion step, which greatly reduces the density of the streamlines produced from the raw DTMRI data. While necessary for fabrication, the result is that the toolpaths manufactured represent mesoscale trends in fiber orientation rather than individual CM cells or fibers. This is still essential information for inclusion in tissue engineering and organ modeling. Beyond fibers, some researchers have reported the presence of sheetlet structures—layers of 5–10 tightly grouped CM fibers—in the myocardium based on DTMRI analysis[37,39]. While not universally acknowledged, it has been proposed that sheetlets play a role in ventricle wall thickening during systole and may be altered in certain disease states[39,40]. In its present incarnation, NAATIV3 is not designed to account for sheetlet structures, which may impact myocardial contraction dynamics in a cellularized version of the model.

Creating exogenous organs models for surgical planning or, eventually, cellularized organs for transplantation requires manufacturing tissues at human scale, as captured by the full-size, 1:1 scale left ventricle model. This model was successfully 3D printed and cured in a FRESH support bath. However, the model lacked stability upon extraction from the FRESH bath. This is potentially due to a lower-density region of toolpaths in the midwall of the 1:1 scale model, which likely originated during the interference removal steps of G-code generation in NAATIV3. Adjusting certain NAATIV3 parameters, such as maximum tracking length or needle diameter, may be able to optimize the density of the full-scale model and enhance its stability after extraction from the FRESH support.

Other considerations for further development of NAATIV3 may include alternative methods of cycle removal to the iterative stochastic optimization method used here. This study explored a heuristic method for acyclic subgraph selection, which was outperformed by the selected method;

however, this does not preclude the existence of a superior heuristic method with lower computational costs than the current stochastic method. The computational resources required to successfully execute NAATIV3's processing steps may constitute a limitation of this framework. This is particularly relevant for data at a 1:1 human scale. Future work to streamline the execution of memory-intensive functions may make NAATIV3 more broadly applicable and accessible. Finally, future work should include demonstrating the use of NAATIV3 on a broader data set. This paper presented tractography data but no downstream processing from a human heart with a history of myocardial infarction. While it is thought that a diseased cardiac data set, and theoretically data from any other fibered tissue or structure, would be compatible with the framework, this has not been empirically demonstrated.

This paper presents just one example of the capabilities of NAATIV3. The NAATIV3 framework was designed to be compatible with any fibrous material, and future work may prove NAATIV3 highly effective for biological applications ranging from fabricating non-cardiac organ models to bioprinting plant structures and even to creating lab-grown food[40–42]. NAATIV3 also represents a paradigm shift in field-guided toolpath generation, and may find uses outside of biomedical engineering, such as in principal stress field-guided 3D printing for optimized part strength. NAATIV3 makes it possible for the first time to fabricate three-dimensional constructs that incorporate native fiber patterns, thus recreating the structure–function relationships that are vital to biology and beyond.

## Methods

### DTMRI scan acquisition
DTMRI scans were performed on fixed, ex vivo human heart specimens as described previously by Eggen[43]. The study protocol was reviewed and approved by the Human Subjects Committee of the Institutional Review Board at the University of Minnesota. Briefly, heart specimens were placed in a plastic container and fully submerged in formalin. They were imaged using a 3 T MAGNETOM Trio scanner (Siemens, USA) at room temperature. Diffusion-weighted images were acquired in the ventricular short axis orientation with a voxel size of $1.875 \times 1.875 \times 4.0$ mm. Scan acquisition parameters for the heart with a history of myocardial infarction were as follows: echo time = 98 ms; pulse repetition time = 4000 ms; slice thickness = 4 mm; slice spacing = 4 mm; $b$ value = 1100 s/mm$^2$; number of diffusion gradient directions = 6; number of averages = 1; matrix size = $128 \times 128$. Scan acquisition parameters for the healthy heart were as follows: echo time = 93 ms; pulse repetition time = 4008 ms; slice thickness = 4 mm; slice spacing = 4 mm; $b$ value = 1200 s/mm$^2$; number of diffusion gradient directions = 6; number of averages = 1; matrix size = $128 \times 128$. Both hearts also included a null-weighted image ($b = 0$ s/mm$^2$).

### DTMRI data conversion and preprocessing
Raw DICOM files from the scanner were converted to NIFTI format using the dcm2nii software package[44]. Once converted, the NIFTI file, $b$ value text file, and b vector text file were loaded into Python. The function patch2self from the Diffusion Imaging in Python (DIPY) library was used to remove noise from the raw data[45]. To confirm that the denoising was successful, a slice of the DTMRI scan can be visualized before and after denoising, along with the residual plot, which is defined as:

$$\sqrt{2\left(X_{Original} - X_{Denoised}\right)} \tag{1}$$

where $X$ represents the entire data set. This plot should appear as random noise in a scan where denoising was successful (Supplementary Fig. 6).

To create isotropic voxels, the data were resampled to have $0.8 \times 0.8 \times 0.8$ mm voxels. This was done both to increase data density and to comply with standard tractography procedures which require isotropic input data for interpolation of the primary diffusion direction in each voxel. The data were scaled by a factor of 0.25 along each axis to yield a 1:4 scale model with

isotropic voxel dimensions $0.2 \times 0.2 \times 0.2$ mm. Toolpaths were created using NAATIV3 for both the 1:1 scale model and the 1:4 scale model.

## Left ventricle mask generation

A mask of the LV was created using the manual selection tool in ITK Snap[46]. Image data was viewed in the short axis view and the LV myocardium was visually separated from the background and the right ventricle (Supplementary Fig. 1B). The entirety of the intraventricular septum was included in the mask. Papillary muscle structures were excluded from the mask to create a smooth-walled ventricle model. Valvular structures were also excluded, as they are not part of the myocardium. The mask was exported in NIFTI format and imported into Python.

## Tractography of the LV myocardium

Tractography was executed using a modified version of the Euler Delta Crossings (EuDX) method from the open-source DIPY library[18]. A 4th order, Runge-Kutta numerical integration procedure was implemented and substituted for the existing 1st order Euler's method. All further NAATIV3 steps were implemented in an in-house-developed Python/C code suite. See Supplementary Methods 2.3 for full tractography details.

The tractography step begins by generating a uniform set of seed points throughout the input bounding volume at a density $n_{seed}$. Starting from each seed point, each streamline is propagated through space with a step size of $\Delta s$ along the input vector field, terminating upon reaching a predefined maximum length $l_{max}$ or the edge of the bounding volume. The sign of the input vector field is arbitrary, such that each seed point is initially propagated in the forward and backward direction, yielding two distinct streamlines for each seed point. The sign of the vector field at each propagation step is determined such that directionality is retained.

The input vector field was obtained by taking the continuous trilinear interpolation of the per-voxel principal eigenvector of the per-voxel diffusion tensor, fit to the DTMRI scan. Seed points for tractography were generated at a uniform spatial density $n_{seed}$ of 1 seed per voxel for the 1:4 scale model and 64 seeds per voxel for the 1:1 scale model. The step size $\Delta s$ was set to 0.125 mm for the 1:4 scale model and 0.5 mm for the 1:1 scale model. The maximum tract length $l_{max}$ was set to 18.75 mm for the 1:4 scale model and 75 mm for the 1:1 scale model.

## Streamline thinning via sweep exclusion

A novel method termed sweep exclusion reduces the tractography-created, initial streamline set to a subset possessing uniform spatial density and a uniform, desired inter-streamline distance. The reduced streamline set is to be used directly as toolpaths for 3D printing, after further processing steps. See Supplementary Methods 2.4 for full sweep exclusion details.

Starting with the longest streamline in the set created via tractography, a bounding cylinder is created by sweeping a circular cross-section along the streamline. See Supplementary Methods 2.4.4 for more details regarding initial path selection. Any other streamlines in the set that intersect the bounding cylinder at any point are removed. After sweeping a streamline, the nearest remaining streamline is selected to sweep next, and the process is repeated until every streamline in the set has either been swept or removed. The radius of the swept cylinder is set to the ideal print line spacing $w_S$. To achieve optimal spacing (print lines that are tangential but not overlapping) $w_S$ should be set to the expected print line diameter.

Minimum Average Direct-Flip (MDF) distance is used to guide subsequent swept path selections[47]. After each sweeping iteration, the remaining path with the smallest MDF distance to the previously swept path is selected. MDF distance is the average pairwise Euclidean distance between two streamlines, which is minimized between streamlines that are parallel, of similar length, and spatially nearby. Effectively, the nearest remaining streamline is swept next, ensuring that the inter-streamline spacing in the reduced set is minimized and therefore as close to $w_S$ as possible.

The tractography-created streamline sets were run though sweep exclusion with an optimal spacing of 0.84 mm for the 1:4 scale model and 1.0 mm for the 1:1 scale model. These values were determined

experimentally by measuring the print line diameter resulting from the physical printing setup used for each model.

## Predicting toolpath interference

A volume-intersection based toolpath interference determination model is used to predict nozzle/material interference between each pair of toolpaths in the thinned set and guide toolpath ordering for printing. A directed dependency graph is created that denotes relative ordering restrictions. An initially empty graph with vertices corresponding one-to-one to toolpaths in the thinned set is created. To populate the graph, every pair of toolpaths is run through the volumetric interference prediction model, and if interference is predicted, the corresponding edge is added to the graph. See Supplementary Methods 2.6 for full dependency graph population details.

A geometric model for the printer's extruder and the deposited material is required to perform the interference prediction. A vertical, axis-aligned, cylindrical printing needle model with diameter $d_n$, equal to the inner diameter of the printing needle, is assumed. The deposited material along each toolpath is defined by the cutting edge of the extruder needle, effectively yielding an elliptical cross section with major axis length $d_p$, which is not necessarily equal to $d_n$. This is to allow for experimental determination of printed line width. If the two constructed volumes intersect at any point, then the nozzle would cut through existing material, resulting in interference. Notably, this interference prediction is directional, such that every pair of toolpaths must be run through the predictive model twice to populate the directed dependency graph.

Directed dependency graphs were populated by setting the nozzle diameter $d_n$ to 0.4 mm for the 1:4 scale model and 0.84 mm for the 1:1 scale model, determined by the physical dimensions of the printing needles used in each case. The material width $d_p$ was set to 0.548 mm for the 1:4 scale model and 1.0 mm for the 1:1 scale model, determined by experimental measurement of printed line width in each case.

## Implementation of a cut plane

An optional cut plane step may be used to reduce the edge density in the constructed dependency graph. Necessity and/or the specific geometry of the cut plane is at user discretion. Specific input vector field and bounding volume geometries may result in large toolpath removal rates in the interference removal procedure, which can be mitigated by splitting toolpaths along a finite plane in space and explicitly prohibiting edges in the dependency graph between toolpaths on either side of the plane. The effect is to remove a large number of edges in the dependency graph, while potentially reducing material adhesion at the plane location. See Supplementary Methods 2.5 for full cut plane details.

The fiber structure of the myocardium is generally helical, such that a large number of inherent cycles exist in the constructed dependency graph. This was mitigated by implementing an angled (near vertical) cut plane through the septum in both the 1:4 and 1:1 scale ventricle models.

## Interference removal

In order to yield an interference-free sequence of toolpaths, there must exist no cycles in the constructed dependency graph. Therefore, the thinned toolpath set is further reduced by selecting a maximum subset of the dependency graph vertices such that the resulting dependency graph of the subset is acyclic. A two-step procedure is used to do so. First, a subgraph of the dependency graph containing no length-2 cycles is found. Second, an iterative stochastic procedure is used to find a maximum subgraph of the length-2-cycle-free subgraph. See Supplementary Methods 2.7 for full acyclic subgraph selection details.

The first step of acyclic subgraph selection is to iteratively remove vertices (and all adjacent edges) from the dependency graph until the remaining subgraph contains no length-2 cycles. At each iteration, the vertex in the graph with the largest product of in-degree and out-degree that is in a length-2 cycle is removed. The process is repeated until no length-2 cycles remain. The resulting subgraph is then run through an iterative stochastic procedure, yielding the largest acyclic subgraph found. At each

stochastic iteration, vertices (and all adjacent edges) are removed at random from the graph until the reduced graph is acyclic. This process is repeated for $N$ iterations, and the largest acyclic subgraph is kept. The toolpaths corresponding to the remaining vertices in the acyclic graph represent the final toolpath set to be ordered for 3D printing.

Alternative acyclic subgraph selection techniques were also developed and tested, including a heuristic-based method (see Supplementary Methods 2.7.1). However, the performance of these methods was found to be far inferior to that of iterative stochastic optimization (Supplementary Fig. 15), which was ultimately adopted into the NAATIV3 framework.

## Toolpath ordering for 3D printing
A greedy search algorithm is used to order the acyclic toolpath set into an interference-free toolpath sequence, to be converted to G-code (or another acceptable file format) for 3D printing. The state of each node in the search tree consists of an unordered toolpath set (initially the entire acyclic toolpath set) and an ordered toolpath sequence (initially an empty sequence). The action space at each node consists of the remaining unordered toolpath set at that node. Each action corresponds to removing one toolpath from the unordered set and appending it to the ordered set. The action space is restricted to only those toolpaths that do not imply interference in the toolpath sequence, according to the interference-predictive dependency graph. Nodes are traversed based on a heuristic function possessing two properties: first, the heuristic cost of each parent node is necessarily greater than the heuristic cost of each of its children nodes. This effectively renders the search a depth-first search. Second, for a set of sibling nodes, the node with the lowest total non-extruding travel distance between toolpaths in its ordered sequence will have the lowest heuristic cost. This minimizes the total travel distance in the solution toolpath sequence found, which minimizes print time. See Supplementary Methods 2.8 for full search algorithm details, including derivation of the heuristic function.

## Preparation of FRESH support material
A gelatin microparticle support bath was prepared as previously described[3][5,36]. Briefly, 32% (volume) deionized water and 68% (volume) 140 proof ethanol were mixed at 45 °C. Gelatin type B (Fisher Scientific, Waltham, MA) and gum Arabic (Fisher Scientific, Waltham, MA) powders were added and stirred for 10 min. The solution was cooled to 10 °C in multiple stages by adding ice while continuing to stir. The solution was collected in 50 mL tubes and centrifuged at 1200 RCF for 4 min. After resuspending in cold deionized water, the mixture was centrifuged again at 500 RCF for 5 min. Gelatin microparticles were resuspended in cold PBS and stored in the refrigerator until use.

To create an optically clear support, the microparticle suspension was centrifuged at 1200 RCF for 5 min, resuspended in 50% iodixanol solution (Cosmo Bio, Carlsbad, CA), and centrifuged again at 3500 RCF for 5 min. The gelatin microparticles were resuspended in 60% iodixanol and refrigerated for a minimum of 12 h. Immediately before printing, the microparticle suspension was centrifuged at 3500 RCF for 15 min and the compacted gelatin microparticles were transferred to a glass beaker to be used as a printing chamber.

To extract cured silicone models from the support, the models and encasing gelatin were transferred to a beaker of warm water and stirred gently until the support liquified and the silicone model could be retrieved and dried.

## Preparation of silicone ink for printing
The silicone ink consisted of 75% (w/w) all-purpose white silicone caulk (General Electric, Boston, MA), 20% (w/w) silicone oil (Sigma-Aldrich, St. Louis, MO), and 5% (w/w) PlatSil™ deadener LV (Polytek, Easton, PA). Components were mixed with a spatula and transferred to a printing syringe (Fisnar, Germantown, WI). While in the syringe, the ink was centrifuged at 1000 RCF for 1 min to remove air bubbles. Immediately before printing, a piston (Fisnar, Germantown, WI) was inserted into the barrel and the needle (Fisnar, Germantown, WI) was attached.

## 3D printing of the fibered model
A high-precision Ultimus™ V EFD (Nordson, Westlake, OH) pneumatic controller dispenser was used in combination with an Aerotech (Pittsburgh, PA) nanopositioning stage for controlled spatial material deposition. This system comprises a generic, 3-axis, extrusion-based 3D printing setup. No hardware changes were necessary to enable non-planar or helical toolpath extrusion.

G-code output by NAATIV3 was input into the Aerotech software, A3200 Motion Composer (version 5.06.001). The syringe of silicone ink, prepared as described above, was connected to the pneumatic controller and mounted on the Aerotech. Metal needles (22 GA for the 1:4 scale model, 18 GA for the 1:1 scale model) were lubricated with silicone oil to prevent sticking of the FRESH support and the silicone ink. Ink was dispensed using 450 kPa pneumatic pressure and the needle moved at 3 mm/s while extruding and 10 mm/s while traveling without extruding. Total printing time was approximately 90 min for 1:4 scale models and approximately 7 h for 1:1 scale models.

## Micro-computed tomography
3D-printed silicone ventricle models were scanned using the X3000 scanner (North Star Imaging, Rogers, MN). Samples were scanned while embedded in the FRESH support material. No modifications to the support or the ink were made in the samples that were scanned. Scans were taken at 115 kV and 430 µA. The distance from the source to the sample was 237 mm. The distance from the scanner to the detector was 682 mm. Scan resolution was approximately $45 \times 45 \times 45$ µm. The equipment was calibrated after every scan.

## Micro CT segmentation
Micro CT scan data were segmented to include only the extruded silicone using Materialize Mimics (Version 26.0, Materialize, Belgium). The silicone was able to be differentiated from the background and FRESH by pixel intensity. Thresholding minimums were set at ~80 GV and thresholding maximums were set at ~110 GV. Due to the similarities in color and texture between the beaker and silicone on the micro-CT scan, selection of the silicone was prioritized during thresholding and the beaker was manually removed from the segmentation. The final segmentation results were exported as an STL file.

## Registration of micro-CT and G-code STL
The micro-CT STL was compared to a second STL created from the G-code toolpath sequence, where each toolpath was rendered cylindrically with the diameter $d_p$, the expected diameter of the extruded material. Using the 3D-printed alignment box in the micro-CT STL, a planar surface for each side of the box was generated. The intersections of the planes were used to define the eight corner points of the box. Coordinates for the eight coordinates were taken directly from the G-code file. Corner coordinates from each model were ordered such that they were 1:1 mapped (i.e., the top, front, right corner of the micro-CT box was in the same index as the top, front, right corner of the G-code point). The Procrustes Method was used to compute the transform matrix between the micro-CT and G-code box coordinates, and was then applied to the micro-CT STL to align the two geometries[48].

## Model to model distance analysis
Aligned model geometries were loaded into 3D Slicer software (www.slicer.org). The Model to Model Distance module was used to calculate the signed distance between the source (micro-CT) and target (G-code) surfaces. Model offset was visualized on the micro-CT model using built-in colormap options. Calculated distances were exported as a.csv file. Python was used to compute the mean and standard deviation of the distances.

## Data availability
DTMRI data can be made available upon reasonable request. All other data generated in this study can be made available upon reasonable request to the corresponding author.

## Code Availability

Source code (Python and C) and sample data for the NAATIV3 framework are available via an academic and non-commercial use license from the University of Minnesota Technology Commercialization office. The license can be accessed here: https://license.umn.edu/product/3d-vector-field-guided-toolpathing-for-3d-bioprinting.

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

## Acknowledgements

The authors thank the Visible Heart Laboratory, in particular R. Brigham, N. Duong, and J. Brigham, at the University of Minnesota for providing DTMRI data and assisting with micro-CT scans; LifeSource (Minneapolis, MN) for facilitating the donation of human organs for this research, as well as the generous gifts of the donors and their families. M.R.G. was supported by the NIH through the Cardiovascular Engineering Training Program at the University of Minnesota (T32-HL139431) and an NSF Graduate Research Fellowship (2237827). N.P.R. was supported by the NIH through the Cardiovascular Engineering Training Program at the University of Minnesota (T32-HL139431).

## Author contributions

M.R.G. conceived the study. M.R.G. and S.E.B. designed the experiments. S.E.B. performed the computation. M.R.G. and N.P.R. performed physical experiments. M.R.G. analyzed and interpreted the data. R.J. contributed technical advice and ideas. M.R.G. and S.E.B. wrote the manuscript. A.P-M. advised on the writing and formatting of the manuscript. M.C.M. oversaw the study. All authors revised the manuscript.

## Competing interests

M.R.G., S.E.B., R.J., A.P-M., and M.C.M. have filed patent application US 18/769,027 "Directional Imaging Based Toolpath Determination." The other authors declare no competing interests.
