## [Transparent Peer Review file · Communications Engineering]

3D vector field-guided toolpathing for 3D bioprinting

Corresponding Author: Professor Michael McAlpine

Version 0:

Reviewer comments:

Reviewer #1

(Remarks to the Author)

The cardiomyocytes in the heart form a 3D continuum with a highly anisotropic microstructure. The paper submitted by Griffin et al describes a new approach to attempt to recreate this structure using 3D bioprinting. As the authors correctly point out most current bioprinting techniques either do not take the anisotropic structure of the myocardium into account or replicate it poorly, usually resulting in successive or individual monolayers. Here the authors use a DTI-tractography dataset as a 3D template and print true 3D trajectories using a silicone bioink suspended in a gelatin-like material. This is one of the best, if not the most accurate, representations of the myocardium using 3D bioprinting I have encountered and represents a significant advance. There is much to commend in this paper but several elements could be further clarified and some limitations further discussed.

General Thoughts:

- 1) The images in the paper clearly show that the bioprinting approach produced oblique tracts in the subepicardium. However, it is difficult to visualize any tracts in the midmyocardium and subendocardium in any of the figures since these are naturally obscured by the subepicardial tracts. Please provide separate images of the following, either via segmentation of the micro-CT images or by successive removal/cutting of layers of the bioprinted heart.
 - a. Subepicardial tracts (oblique, negative helix angle)
 - b. Midmyocardial tracts (circumferential, zero helix angle)
 - c. Subendocardial tracts (oblique, positive helix angle).
- 2) The microstructure of the heart occurs at two levels. The cardiomyocytes are arranged in a crossing helical pattern (resolved with the primary eigenvector or helix angle) and groups of cardiomyocytes are arranged into sheetlets (resolved with the secondary eigenvector or sheetlet angle). While the authors have made a significant advance in accurately bioprinting the helical arrangement of the cardiomyocytes, their model does not replicate the sheetlet structure of the myocardium at all. This will likely be very challenging and beyond the scope of this paper. However, the authors should acknowledge and discuss this limitation in the discussion because, as discussed briefly below, the sheetlet structure of the heart is central to its ability to contract/thicken.
 - a. As the authors correctly point out the helix angle of the cardiomyocytes in the heart ranges from +60 to -60 degrees. There are no radially oriented cardiomyocytes in the heart, which creates a paradox since the primary eigenvector of myocardial strain is radial. This radial strain is produced by sheetlet rotation, shearing and extension during systole (in the DTI literature described by an increase in E2A in systole - see for example NIELLES-VALLESPIN et al JACC 2017), hence the importance of sheetlet architecture.
- 3) The sheetlets of the heart contain 5-10 tightly grouped cardiomyocytes but there are gaps between adjacent sheets. The spacing between cardiomyocytes in the heart is therefore not uniform as it is modelled here with a uniform W_s .
- 4) The cardiomyocyte streamlines modeled with the bioink were 0.54mm or 1mm wide (1:1 vs 1:4 modeling). The human heart is 8-10mm thick (in diastole) and this would result in some averaging of several physical trajectories and create an average trajectory in each streamline. This is not a major problem, and a similar phenomenon occurs with DTI of the heart, but it should be acknowledged.
- 5) DTI-tractography data are highly interpolated. The authors downsample or thin these oversampled/interpolated datasets to create a feasible 3D field for bioprinting. One wonders if the tractography was performed with a significantly lower degree of interpolation whether the sweep exclusion step the authors describe would be needed at all.
- 6) The authors introduce a "cut-plane" into the heart to identify and eliminate cyclic loops. As they point out this can be problematic with helical structures, and the circumferentially oriented tracts in the midmyocardium do have the potential of forming repetitive loops on DTI. However, could this cut plane introduce an artificial discontinuity into the myocardial continuum and reduce the physiological accuracy of the model?

Figures:

- 1) Perhaps I am missing something, but I could not find a Figure 1 in the main manuscript?
- 2) Figure 2. The DTI maps show foci in the myocardium with helix angles greater than +/-70 degrees. These foci are almost certainly trabeculations and should be labeled/identified as such.
- 3) Figure 2E and Figure 3E– Please provide some anatomical labels to indicate the orientation of the heart. This is important since the tracts in the lateral wall and septum have the same helix angle distribution but opposite orientations. For instance, subendocardial tracts in the lateral wall (positive helix angle) align from posterobase to anteroapex and the subendocardial tracts in the septum (identical positive helix angle) from anterobase to posteroapex. This orientation is inherent to forming a physical helix but cannot be easily visualized in any of the figures (see general comments above)
- 4) Fig 5 – The images show very nice oblique tracts in the subepicardium but their orientation (see point 3) is not well enough depicted. In addition, the tracts in the midmyocardium and subendocardium cannot be visualized.

Methods:

- 1) Lmax (tract length) was set 18.7mm and 75mm (1:1 vs 1:4). How was this tract length selected? The selected Lmax would be well below the circumference of even a small human heart. What are the implications of this?
- 2) The step size (delta S) is listed as 1.25 or 5mm in the supplement and 0.125 or 0.5mm in the methods. Please clarify. Also, if a step size of 5mm was used this is very high.
- 3) The acquired resolution of the DTI images was 1.87x1.87x4mm. An isotropic voxel of 2mm would produce only slightly lower SNR and the longer scan time (doubled) should not be an issue in a fixed heart. Why was this anisotropic resolution chosen?
- 4) The acquired images were interpolated to 0.8mm isotropic resolution. In-plane this is a factor of 2.3, but through plane this is a factor of 5x interpolation in resolution. This seems very high.
- 5) Please list the number of diffusion encoding directions used (6, 12, 32.....).
- 6) A tract termination angle of 35 degrees is generally used for DTI-tractography of the heart, although this is far higher than is needed in a healthy heart. What tract termination angle was used in the bioprinted streamlines?
- 7) What was the Young's modulus of the silicone bioink? Is it similar to the Young's modulus of the myocardium?

Reviewer #2

(Remarks to the Author)

Reviewer:

This work presents a framework named NAATIV3 that processes DTMRI data to map tissue fibers, optimizes them to a representative subset, eliminates conflicting fibers, determines the optimal printing sequence, and generates a G-code file. Using DTMRI data from a human left ventricle, this allowed the authors to successfully 3D printed fibered models with good accuracy.

In particular, this work outline a method for streamlining tractography data for 3D bioprinting: here the work could benefit from addressing how the method handles non-parallel fiber orientations, the impact of scaling, and structural preservation post-thinning.

The authors proposed a structured and logical method for generating non-interfering toolpaths during 3D bioprinting (actually 3D printing in this work), leveraging a dependency graph.

Moreover, it is offered a systematic approach to ensuring that the set of 3D bioprinting toolpaths is acyclic (needed to avoid material interference during printing).

The method for generating and printing toolpaths is eventually successfully demonstrated in its effectiveness for printability and dimensional accuracy.

The text is technically sound and presents a clear methodology for adapting dense streamline data from tractography to extrusion-based 3D bioprinting. However, some major points need to be addressed and overall, the text could benefit from a more substantial discussion to give a fair context to the achieved results: some further elaboration on the limitations of this technique is needed. The subject treated is very relevant to the tissue engineering field and the method has novelty and utility potential, with perspective substantial technological impact. Thus, I recommend this text for publication upon major revision, to address the following points.

1. In the abstract, the authors state: "To date, there are no reports of directly incorporating this alignment information into engineered tissue."

More words and precision are needed here to avoid misinterpretations or unfair claims. This claim would be likely inaccurate or outdated because: (i) over the past decade, there has been significant progress in tissue engineering, particularly in replicating fiber alignment, (ii) techniques like electrospinning, 3D bioprinting, and magnetic alignment have been used to engineer tissues with controlled fiber orientation, (iii) studies have explored using DTMRI data to design scaffolds that mimic native tissue architecture, especially in muscle and neural tissue engineering. I would suggest a rephrasing, maybe along the lines of: "While directional imaging modalities such as diffusion tensor magnetic resonance imaging (DTMRI) can derive a 3D vector map of fiber orientation, incorporating this alignment information into engineered tissues remains a challenging and evolving area of research."

2. Major comment on the chosen applicative target (fiber alignment within tissues): it is generally accepted that 3D bioprinting using extrusion-based methods often struggles to provide precise cell alignment at the micro or nanoscale, which is crucial for replicating the natural organization of tissues, particularly muscle tissue. Muscle cells, such as myocytes, require specific microstructural cues to guide their alignment and orientation, or they respond to mechanical stimuli to properly align with one direction. These physical cues can include features like grooves and topographical patterns, which

are often smaller than the resolution provided by typical extrusion-based bioprinting (micro-scale range). The resolution of bioinks used in extrusion printing typically ranges from hundreds of microns (typically starting from 300 μm up to millimeters), thus larger than the microstructural organization required for muscle fibers to align and function properly. As a result, while extrusion-based bioprinting can successfully deposit cells and extracellular matrix components in three dimensions, this method - alone - often fails to recreate the necessary fine-grained organization to direct the cells into functional, aligned patterns. Thus, it is expected that, despite the successful deposition, a cardiomyocyte bioink deposited with the proposed method will not generate aligned cells according to the pattern provided. Rather, the cells will randomly distribute within each deposited ink line.

It is suggested that the authors discuss the relevance of their method more to replicate "meso" scale level of the tissue architecture, rather than pointing out directly to cell alignment, as this could be misleading. This is very crucial for the cardiac muscle, in which the cellular alignment is crucial to determine the crucial motile function of the tissue.

I believe that the discussion of the manuscript is in its present form quite limited and could benefit from further elaboration. This point could be an example of additional elaboration.

3. A major point in this paper is the lack of complete actual tissue biofabrication as a proof of concept. As said in the point above, even if the demonstration of a biofabricated heart tissue would have been provided, it is unlikely that the tissue could really produce aligned myofibers. Instead the potential result could be a cluster of mm-scale line-like tissues, organized in oriented arrays as seen in the demonstration with synthetic inks. This structure would still certainly be a relevant achievement, but still requires a better contextualization, especially in the absence of the proof of principle of an actual heart tissue achieved. This reviewer understands the difficulty of such realization and believes that the realization based on synthetic inks is sufficient to support the message of the paper. However, as said in the points above, it is recommended that the authors explain the limitations of the current study in a more complete manner, to avoid overclaiming in relation to the chosen application (cardiac tissue biofabrication).

4. Beginning of the introduction: the idea that "relatively little progress has been made" in recapitulating fiber microarchitecture could be seen as an overstatement. Some methods, although not perfect, do exist and have shown promise.

5. "Directional data is voxelized, where each voxel is a uniform, cubic volume that contains a subset of the data": is this a regular result of voxelization? Not all voxelization processes necessarily result in cubic volumes, depending on the imaging modality. For a broader audience, it could be beneficial to add a small addition about why choosing cubic voxels and how this choice - or these cubic voxel - interact with fiber orientation data or affect them.

6. "...custom heuristic function for the greedy search procedure": if possible, the authors might want to anticipate at this point what metrics or factors are considered in this heuristic. While the utility in printing time reduction is clear, maybe this information could make the statement more complete/convincing.

7. The authors mention that further mathematical and computational details are in the Supplementary Methods. However, as for the points above, the main text could benefit from a brief outline of the core mathematical principles guiding the NAATIV3 framework (especially concerning vector field processing and toolpath optimization) at this early text point as well. This would result in increasing clarity and completeness.

8. As another idea to make the discussion more complete and comprehensive, I propose the author considering the following point: is there any data or perspective about how NAATIV3 can address or mitigate common tractography errors? Tractography, especially if extracted from DTMRI, is affected by inaccuracies (example, where there is fiber crossing, bending, branching or other tortuosities...). The authors could discuss potential limitations or validation techniques for the accuracy of the generated streamlines.

9. "Resampling to a 1:4 scale for the purpose of an in vitro model": how can this scaling affect the biomechanical properties or the fidelity of fiber alignment when translated to a bioprinted construct? Is it possible that changes in scale can alter the geometric and mechanical characteristics? If so, this might impact the interpretation of the results? For example, when the authors discuss using different gauge needles (22 GA in 1:4; and 18 GA in the 1:1 scale), different actual material diameters are reported on data obtained via micro-CT. Maybe, one could mention that the differences in printed filament thickness are expected to impact the biomechanical properties or alignment fidelity of the final construct, especially for scaling up from a model to a realistic tissue size.

10. To eliminate overlapping streamlines, using a bounding cylinder may not be the optimal solution as it does not fully account for complex fiber curvatures (critical when fibers cross or twist). So, how does the algorithm handle situations where the curvature of the fiber significantly deviates from linearity within a local region? Is this going to affect the accuracy of the final toolpath set?

11. Can the authors explain if preferring the longest streamline first would generate inaccuracy in replication? Namely, can this bias the structure toward certain spatial regions? Is there any risk to potentially obtain uneven toolpath distribution (example: in areas where fiber lengths naturally vary a lot)?

12. "structural validity of the set is maintained" after thinning: is this validated in any quantitative metrics or specific methods? Could the authors make this claim more robust by including quantitative measures demonstrating fiber/streamlines patterns preservation? Would it be possible to compare angle distributions before and after thinning, for example?

13. Another idea to substantiate the discussion of these results could be elaborate on the advantages and disadvantages of prioritizing longer streamlines to increase toolpath continuity. This is surely reasonable and intelligent, but could this not necessarily result in always the optimal path for minimizing print time or material use? Is there any perspective on more nuanced approaches for example, balancing length with spatial distribution?

14. Using the inner diameter rather than the outer diameter (OD) should reduce the number of calculated dependencies. Will this not result in underestimating collision events/risks? In complex/ densely packed structures (like myocardial tissue), could such a approximation result in unintended overlaps? Thus leading to structural defects?

15. The choice of the cut plane is user-defined, which introduces subjectivity and variability. On this point, the authors can further elaborate in the discussion section. Would it be desirable in the future to find an automated optimization algorithm to determine the optimal plane location, based on fiber orientation analysis rather than manual input?

16. It seems that the iterative stochastic optimization for the largest acyclic subgraph is computationally expensive. Would it be desirable to attempt heuristic methods (or graph simplification?) to reduce the computational load. ML could also predict problematic cycles if graph topology is given. Again, this problem could be emphasized in the discussion.

17. Several toolpaths in the 1:1 model ended up to be removed: may this compromise the accuracy or fidelity of the final printed construct? Is the loss of these paths going to affect the mechanical or functional properties of the model?

18. "... incur minimal impact on the integrity of the final construct". Can the author point out to the reader which figure or data or reference validate this statement. I suggest the author substantiate this claims with some observations or comments to the related figure.

19. In the Discussion: "By transcending the planar paradigm and using native fiber tracts to inform toolpath generation, it is possible to 3D bioprint tissue models with a near-native fiber structure for in vitro research, preclinical modeling, and surgical planning." This statements sounds to me as an overclaim, as in this paper no actual 3D bioprinting of tissue model took place, and "near-native fiber structure" as referred to muscle tissue is misleading. "Fiber" is typically the myofiber, which is a structure way below the resolution of 3Dbioprinting (see point 2).

I suggest rephrasing this sentence for fair claim and clarity on the actual achievement of this paper. The concept here expressed can also be kept but requires re-elaboration in terms of perspective (e.g., "hold potential for.." or similar phrasing).

20. Line 309- discussion: "...inability to realistically capture native fiber structure", same comments as above.. I suggest the authors refers to mesoscale architecture of the tissue.

21. The claim that "any bioink could be substituted" lacks experimental validation. The results are obtained with silicone-based ink and so these results may not directly translate to cell-laden bioinks (more prone to shear stress damage and settling in the FRESH bath). If the author will not provide validation data concerning the statements related to "actual bioprinting conditions", then it would be better to elaborate on the current lack of experimental data and open perspectives for future works which could actually prove the technique validity in a real tissue engineering work.

22. The reported deviation is 0.138 ± 0.250 mm, and the max. dev. is 3.9mm. Is this correct? This significantly exceeds the width of the extruded line (0.54 mm). This max deviation looks to be substantial. Would this compromise the anatomical fidelity of the printed model? If so, how can this be potentially corrected?

23. Authors claim successful printability and shape accuracy of the printed model, but I could not find any mechanical or functional tests to evaluate the integrity of the printed structure. If so, then the concern would be that the model's mechanical strength, flexibility, and fiber orientation accuracy are crucial, particularly when aiming to replicate biological tissue properties. Verifying geometric accuracy alone is insufficient for applications such as surgical planning or functional tissue engineering. As for the biological validation (if the goal is to use cell-laden bioinks), mechanical testing (e.g., tensile strength, compressive modulus) to assess whether the printed model behaves similarly to native tissue would be recommended. However, as this could go beyond the scope of the work, it is recommended that the authors use this point to better discuss the limitations of their achievements and provide realistic/pragmatic outlooks on the next steps in this research.

Version 1:

Reviewer comments:

Reviewer #1

(Remarks to the Author)

The authors have addressed most of my concerns and the revised paper, particularly Figure 5, is much improved.

Reviewer #2

(Remarks to the Author)

I would like to thank the authors for their thoughtful and thorough revisions. I am very pleased with the changes applied to the manuscript, which I believe have significantly improved the clarity, structure, and overall quality of the work. The authors addressed my comments with impressive accuracy and care, demonstrating both scientific rigor and attention to detail. I consider the manuscript ready for publication in its current form.

Additional remarks: I appreciate this work, whose major claim is the development and demonstration of the NAATIV3 framework, which enables the translation of DTMRI fiber orientation data into executable toolpaths for 3D bioprinting. The authors unlock the fabrication of tissue-engineered constructs with biologically relevant fiber alignments, directly informed by native tissue architecture. This is a novel and timely contribution to the field of biofabrication, as existing approaches often rely on indirect, heuristic, or simplified representations of anisotropy rather than direct mapping from imaging data. The main value in publicizing this article will be providing the readers with a tool that finally bridges the gap between directional imaging and 3D printing. This method not only adds a new computational tool but also opens up opportunities for higher-fidelity modeling of tissue structure-function relationships. The claim is strongly supported by a well-structured algorithmic pipeline.

The work is convincing, and it is likely to influence thinking in both the bioprinting and computational tissue engineering communities. The authors provide compelling evidence of the accuracy and generalizability of their method, and their (now extended) discussion adds to the significance. The research fills a clear gap in current methodologies, and its interdisciplinary nature will likely attract interest beyond the immediate field.

The authors provided an interesting contribution in the field of accelerating biofabrication, where an urgent need is present. The integration of high-resolution imaging data with bioprinting toolpath generation is still in its early stages, and this paper provides a concrete, scalable solution. I believe it will stimulate further innovations at the interface of imaging, modeling, and fabrication, and contribute meaningfully to the development of functionally and structurally realistic engineered tissues.

Reviewer #1

General comment 1: The images in the paper clearly show that the bioprinting approach produced oblique tracts in the subepicardium. However, it is difficult to visualize any tracts in the midmyocardium and subendocardium in any of the figures since these are naturally obscured by the subepicardial tracts. Please provide separate images of the following, either via segmentation of the micro-CT images or by successive removal/cutting of layers of the bioprinted heart.

- a. Subepicardial tracts (oblique, negative helix angle)
- b. Midmyocardial tracts (circumferential, zero helix angle)
- c. Subendocardial tracts (oblique, positive helix angle).

Response:

We appreciate the reviewer's point about the difficulty of visualizing the deeper regions of our left ventricle model. As suggested, we have segmented the micro-CT data to show the subepicardial, midmyocardial, and subendocardial tracts in our model. These images have been added as Fig. 5 C-E.

General comment 2: The microstructure of the heart occurs at two levels. The cardiomyocytes are arranged in a crossing helical pattern (resolved with the primary eigenvector or helix angle) and groups of cardiomyocytes are arranged into sheetlets (resolved with the secondary eigenvector or sheetlet angle). While the authors have made a significant advance in accurately bioprinting the helical arrangement of the cardiomyocytes, their model does not replicate the sheetlet structure of the myocardium at all. This will likely be very challenging and beyond the scope of this paper. However, the authors should acknowledge and discuss this limitation in the discussion because, as discussed briefly below, the sheetlet structure of the heart is central to its ability to contract/thicken.

- a. As the authors correctly point out the helix angle of the cardiomyocytes in the heart ranges from +60 to -60 degrees. There are no radially oriented cardiomyocytes in the heart, which creates a paradox since the primary eigenvector of myocardial strain is radial. This radial strain is produced by sheetlet rotation, shearing and extension during systole (in the DTI literature described by an increase in E2A in systole - see for example Nielles-Vallespin et al JACC 2017), hence the importance of sheetlet architecture.

Response:

We thank the reviewer for reminding us of the importance of sheetlets in the myocardium. As the reviewer correctly states, including sheetlet architecture in NAATIV3 is beyond the scope of this study. However, the reviewer is also correct that they should not go unmentioned, both because they are also of interest in DTMRI analysis and because they may be implicated in disease states (also discussed by Nielles-Vallespin 2017). We have added a paragraph in the discussion section to address this limitation (lines 361-366).

General comment 3: The sheetlets of the heart contain 5-10 tightly grouped cardiomyocytes but there are gaps between adjacent sheets. The spacing between cardiomyocytes in the heart is therefore not uniform as it is modelled here with a uniform Ws.

Response:

This comment is well aligned with General Comment 2 from the reviewer. Please see our response above.

General comment 4: The cardiomyocyte streamlines modeled with the bioink were 0.54mm or 1mm wide (1:1 vs 1:4 modeling). The human heart is 8-10mm thick (in diastole) and this would result in some averaging of several physical trajectories and create an average trajectory in each streamline. This is not a major problem, and a similar phenomenon occurs with DTI of the heart, but it should be acknowledged.

Response:

The reviewer's point is well taken. We have added language to this effect in lines 161-162.

General comment 5: DTI-tractography data are highly interpolated. The authors downsample or thin these oversampled/interpolated datasets to create a feasible 3D field for bioprinting. One wonders if the tractography was performed with a significantly lower degree of interpolation whether the sweep exclusion step the authors describe would be needed at all.

Response:

The reviewer raises an important question regarding the high interpolation of the initial dataset and the subsequent computationally-intensive sweep exclusion step to reduce the density of this data. This approach is a fundamental component of our procedure. The architecture of three dimensional tractography data is highly complex. The final tract set that is fed to the 3D bioprinter for fabrication relies on proximity between adjacent print lines (tracts) within a tight tolerance range. Neighboring print lines must be sufficiently near each other to adhere to one another - yielding a contiguous printed sample - without being excessively near such that material deposition becomes interfering and disordered. The core functionality of the sweep exclusion step (in particular, when given a highly dense tract set as input) is to reduce the tract set such that the remaining tracts have optimal proximity to each of their neighbors, yielding a final tract set with spatially uniform material deposition and contiguous adherence throughout the sample. If the data was not highly interpolated and the sweep exclusion step was not used, the printed model may possess a comparable overall material density to one printed with our method. However, the printed tracts would be irregularly spaced - highly overlapping in some regions and completely nonexistent in other regions (holes in the model). Such a model would fall well outside the range of tolerances between print lines that is required for a contiguous and well adhering 3D printed model, resulting in a printed sample that would likely struggle to remain intact, and that does not represent the myocardium in its spatially uniform density distribution.

General comment 6: The authors introduce a “cut-plane” into the heart to identify and eliminate cyclic loops. As they point out this can be problematic with helical structures, and the circumferentially oriented tracts in the midmyocardium do have the potential of forming repetitive loops on DTI. However, could this cut plane introduce an artificial discontinuity into the myocardial continuum and reduce the physiological accuracy of the model?

Response:

The reviewer asks a pertinent question. There is some degree of unavoidable discontinuity introduced by adding the cut plane to the model. We have taken two steps to minimize this in our implementation: using a zero-thickness plane and angling the plane. In doing so, we have increased the overlapping surface area of the cut toolpaths and, to the extent possible, created a design where the cut ends will touch during printing. We discuss this in lines 222-224. Viewing the cured model in Supplementary Video 1 and the micro-CT scan in Figure 5B (right), it appears that the touching ends of cut toolpaths have fused after printing, which was our intent.

During the development of NAATIV3, we investigated the impact of removing the cut plane on the model. The final, printable toolpath set was nearly 25% larger with the cut plane than without it, underscoring for us the utility of this method. Given the evidence in its favor, and the steps taken to minimize impacts on physiological accuracy, we believe the tradeoff is justified.

Figure comment 1: Perhaps I am missing something, but I could not find a Figure 1 in the main manuscript?

Response:

We appreciate the reviewer double-checking this for us. Reviewing the original submission, it seems that Figure 1 may have been lost in the upload process, we sincerely apologize for any confusion. The figure is present in the resubmission file.

Figure comment 2: Figure 2. The DTI maps show foci in the myocardium with helix angles greater than +/-70 degrees. These foci are almost certainly trabeculations and should be labeled/identified as such.

Response:

We appreciate the reviewer pointing out this distinction. The Figure 2 caption has been updated to identify the foci above +70 degrees helical angle as trabeculations.

Figure comment 3: Figure 2E and Figure 3E– Please provide some anatomical labels to indicate the orientation of the heart. This is important since the tracts in the lateral wall and septum have the same helix angle distribution but opposite orientations. For instance, subendocardial tracts in the lateral wall (positive helix angle) align from posterobase to anteroapex and the subendocardial tracts in the septum (identical positive helix angle) from anterobase to posteroapex. This orientation is inherent to forming a physical helix but cannot be easily visualized in any of the figures (see general comments above)

Response:

We thank the reviewer for this suggestion. Anatomical orientation references have been added to all relevant figures.

Figure comment 4: Fig 5 – The images show very nice oblique tracts in the subepicardium but their orientation (see point 3) is not well enough depicted. In addition, the tracts in the midmyocardium and subendocardium cannot be visualized.

Response:

Please refer to our response to General Comment #1.

Methods comment 1: Lmax (tract length) was set 18.7mm and 75mms (1:1 vs 1:4). How was this tract length selected? The selected Lmax would be well below the circumference of even a small human heart. What are the implications of this?

Response:

We thank the reviewer for the question. The maximum tract length was selected to be approximately one half the circumference of the human left ventricle, as measured from the DTMRI scan. This length was selected to ensure that helical tracts making more than one complete revolution around the ventricle (which pose problems during the subsequent interference

prediction and removal steps) did not occur, while at the same time providing sufficient tract length such that the final printed model retains contiguity.

Methods comment 2: The step size (delta S) is listed as 1.25 or 5mm in the supplement and 0.125 or 0.5mm in the methods. Please clarify. Also, if a step size of 5mm was used this is very high.

Response:

We appreciate the reviewer catching this typo. The step sizes used are as denoted in the methods: 0.125 and 0.5 mm for the 1:4 and 1:1 scale ventricle models, respectively. The supplement has been updated to correct this error.

Methods comment 3: The acquired resolution of the DTI images was 1.87x1.87x4mm. An isotropic voxel of 2mm would produce only slightly lower SNR and the longer scan time (doubled) should not be an issue in a fixed heart. Why was this anisotropic resolution chosen?

Response:

We appreciate the question from the reviewer. The DTMRI data used in this study was generously shared by a collaborating lab at the University of Minnesota. They selected the voxel size according to their lab's experimental protocols (reported in Eggen 2012).

Methods comment 4: The acquired images were interpolated to 0.8mm isotropic resolution. In-plane this is a factor of 2.3, but through plane this is a factor of 5x interpolation in resolution. This seems very high.

Response:

We thank the reviewer for the question. The data was resampled at a high interpolation factor to: (1) provide a higher-fidelity 3D ventricle mask region, enabling more contiguous tracts to form along the boundary of the ventricle during tractography; and (2) increase the voxel density, enabling a higher seed point density, which in turn yields a higher density of tracts during tractography. Furthermore, trilinear interpolation was used both to resample the original diffusion MRI data to isotropic 0.8 mm voxel size, and to define a continuous vector field from the discrete data for direction sampling during tractography. As such, this voxel resizing had no impact on the tractography results beyond the desired increase in seed point density and mask fidelity. This important feature of linear resampling has been added to Section 2.9 (Cardiac Toolpathing) of the Supplementary Methods.

Methods comment 5: Please list the number of diffusion encoding directions used (6, 12, 32.....).

Response:

We thank the reviewer for pointing out this oversight. The number of diffusion encoding directions used was 6. This information has been added to the 'DTMRI scan acquisition' section of the methods.

Methods comment 6: A tract termination angle of 35 degrees is generally used for DTI-tractography of the heart, although this is far higher than is needed in a healthy heart. What tract termination angle was used in the bioprinted streamlines?

Response:

The reviewer is correct in stating that using a tract termination angle of 35 degrees is a common method employed in myocardial tractography. In our implementation of NAATIV3, we have

substituted the maximum tracking length condition and the edges of the boundary volume as tract termination criteria. This information is included in lines 447-449.

Methods comment 7: What was the Young's modulus of the silicone bioink? Is it similar to the Young's modulus of the myocardium?

Response:

We thank the reviewer for the question. In other research conducted in our lab, we have measured the Young's modulus for the pure silicone component of our bioink to be 0.083 MPa. Brazhkina and colleagues (2021) reported the Young's modulus of the myocardium to be between 0.02 and 0.5 MPa. The reported range is very large because measuring a precise modulus for the myocardium is difficult. Moduli reported from testing bulk sections of the tissue are very different from those reported for fibers isolated from a single layer of muscle.

We decided to focus on developing a silicone bioink with good printability that would fully cure in our aqueous support material. We did not pursue further mechanical testing with our final ink formulation, as we felt it was outside the scope of the work. This is certainly an interesting direction for future research, and there is potential to tune multiple ink formulations to create a range of mechanical properties in synthetic fibered models. We have added this suggestion to the discussion section in lines 348-354.

Reviewer #2

Comment 1: In the abstract, the authors state: “To date, there are no reports of directly incorporating this alignment information into engineered tissue.”

More words and precision are needed here to avoid misinterpretations or unfair claims. This claim would be likely inaccurate or outdated because: (i) over the past decade, there has been significant progress in tissue engineering, particularly in replicating fiber alignment, (ii) techniques like electrospinning, 3D bioprinting, and magnetic alignment have been used to engineer tissues with controlled fiber orientation, (iii) studies have explored using DTMRI data to design scaffolds that mimic native tissue architecture, especially in muscle and neural tissue engineering. I would suggest a rephrasing, maybe along the lines of: “While directional imaging modalities such as diffusion tensor magnetic resonance imaging (DTMRI) can derive a 3D vector map of fiber orientation, incorporating this alignment information into engineered tissues remains a challenging and evolving area of research.”

Response:

We appreciate the reviewer’s suggestion. As the reviewer says, there has been much work undertaken on the topic of replicating fiber architectures in the last decade. Our intent was to convey that the vast majority of reports use a derivation of native fiber orientation when designing their models, rather than directly incorporating fiber orientation information derived from medical imaging or other sources. Even studies that reference DTMRI or other directional imaging as the inspiration for their scaffold designs do not incorporate the use of actual raw data. One example of this is found in Mohammadi (2022).

The abstract language has been updated to clarify this intent and incorporate the reviewer’s suggestion.

Comment 2: Major comment on the chosen applicative target (fiber alignment within tissues): it is generally accepted that 3D bioprinting using extrusion-based methods often struggles to provide precise cell alignment at the micro or nanoscale, which is crucial for replicating the natural organization of tissues, particularly muscle tissue. Muscle cells, such as myocytes, require specific microstructural cues to guide their alignment and orientation, or they respond to mechanical stimuli to properly align with one direction. These physical cues can include features like grooves and topographical patterns, which are often smaller than the resolution provided by typical extrusion-based bioprinting (micro-scale range). The resolution of bioinks used in extrusion printing typically ranges from hundreds of microns (typically starting from 300 um up to millimeters), thus larger than the microstructural organization required for muscle fibers to align and function properly. As a result, while extrusion-based bioprinting can successfully deposit cells and extracellular matrix components in three dimensions, this method - alone - often fails to recreate the necessary fine-grained organization to direct the cells into functional, aligned patterns. Thus, it is expected that, despite the successful deposition, a cardiomyocyte bioink deposited with the proposed method will not generate aligned cells according to the pattern provided. Rather, the cells will randomly distribute within each deposited ink line.

It is suggested that the authors discuss the relevance of their method more to replicate “meso”scale level of the tissue architecture, rather than pointing out directly to cell alignment, as this could be misleading. This is very crucial for the cardiac muscle, in which the cellular alignment is crucial to determine the crucial motile function of the tissue. I believe that the discussion of the manuscript is in its present form quite limited and could benefit from further elaboration. This point could be an example of additional elaboration.

Response:

We appreciate the detailed comment from the reviewer. Cellular alignment during the bioprinting process may stem from shear forces experienced during extrusion, contact guidance cues provided by the edges of the extruded material, or most likely, a combination of the two. Multiple studies have demonstrated that extrusion-based bioprinting may be sufficient to drive cellular alignment. Of particular relevance to this publication is Maiullari (2018), who reported cardiomyocyte alignment within 10 degrees of parallel to the direction of extrusion bioprinting with no additional cues or forces applied. Other studies, including Studart (2016), Mozeti (2017), and Bera (2025), have reported similar results.

The reviewer is correct that we have not yet attempted to replicate these studies in our research group or apply them to our left ventricle model. It was not our intent to claim that we have proven or disproven any theory of cellular alignment. Rather, we meant to demonstrate NAATIV3 in the context of one application – cardiac tissue engineering – that we are actively researching. We have added discussion of this and the suggestion for future research directions related to cardiomyocyte alignment to the discussion section of the manuscript in lines 336-347. We have also added additional comments on the value of higher-level, mesoscale, tissue architectures in the field.

Comment 3: A major point in this paper is the lack of complete actual tissue biofabrication as a proof of concept. As said in the point above, even if the demonstration of a biofabricated heart tissue would have been provided, it is unlikely that the tissue could really produce aligned myofibers. Instead the potential result could be a cluster of mm-scale line-like tissues, organized in oriented arrays as seen in the demonstration with synthetic inks. This structure would still certainly be a relevant achievement, but still requires a better contextualization, especially in the absence of the proof of principle of an actual heart tissue achieved. This reviewer understands the difficulty of such realization and believes that the realization based on synthetic inks is sufficient to support the message of the paper. However, as said in the points above, it is recommended that the authors explain the limitations of the current study in a more complete manner, to avoid overclaiming in relation to the chosen application (cardiac tissue biofabrication).

Response:

We appreciate the reviewer's comment, and we appreciate that they see the value in our synthetic, fibered model of the left ventricle. Please see the response to Comment 2, above, for how we have addressed this point in the discussion section of the manuscript.

Comment 4: Beginning of the introduction: the idea that "relatively little progress has been made" in recapitulating fiber microarchitecture could be seen as an overstatement. Some methods, although not perfect, do exist and have shown promise.

Response:

We apologize for the confusion, we certainly did not mean to overstate our position or undervalue the advancements made in this space by other researchers. Our intent was to capture the limitations discussed in the remainder of that paragraph, namely that current methods are typically uniaxial or confined to planar layers. We have updated the text to reflect the suggestion in the reviewer's Comment #1.

Comment 5: "Directional data is voxelized, where each voxel is a uniform, cubic volume that contains a subset of the data": is this a regular result of voxelization? Not all voxelization processes necessarily result in cubic volumes, depending on the imaging modality. For a broader audience,

it could be beneficial to add a small addition about why choosing cubic voxels and how this choice - or these cubic voxel - interact with fiber orientation data or affect them.

Response:

We appreciate the reviewer pointing out this misstatement. We have corrected the language to remove “cubic” from the text. We comment on the need for cubic voxels in lines 422-424. The shape of the voxels has no impact on the fiber orientation data acquired from the DTMRI images.

Comment 6: “...custom heuristic function for the greedy search procedure”: if possible, the authors might want to anticipate at this point what metrics or factors are considered in this heuristic. While the utility in printing time reduction is clear, maybe this information could make the statement more complete/convincing.

Response:

We appreciate the reviewer’s suggestion to expand further upon the greedy heuristic used in the NAATIV3 framework. The corresponding portion of main text has been updated to further explore the features of this custom heuristic method (lines 114-117).

Comment 7: The authors mention that further mathematical and computational details are in the Supplementary Methods. However, as for the points above, the main text could benefit from a brief outline of the core mathematical principles guiding the NAATIV3 framework (especially concerning vector field processing and toolpath optimization) at this early text point as well. This would result in increasing clarity and completeness.

Response:

We appreciate the reviewer’s proposal. While the main text contains little mathematical formulation with respect to the NAATIV3 framework, further expansion upon the core concepts discussed would use a significant amount of space and go beyond the scope that is interesting to the typical reader. In order to meet length requirements and retain conciseness in the main text, we feel the current main figure set, in conjunction with the available supplementary methods and figures, provide sufficient context for our discussion while retaining broad accessibility to readers from various technical backgrounds.

Comment 8: As another idea to make the discussion more complete and comprehensive, I propose the author considering the following point: is there any data or perspective about how NAATIV3 can addresses or mitigates common tractography errors ? Tractography, especially if extracted from DTMRI, is affected by inaccuracies (example, where there is fiber crossing, bending, branching or other tortuosities..). The authors could discuss potential limitations or validation techniques for the accuracy of the generated streamlines.

Response:

We appreciate the reviewer’s question regarding anatomical inaccuracies stemming from numerical tractography. While mitigating and/or analyzing tractographical error is not a core functionality nor a novel aim of NAATIV3, the implementation of 4th order Runge-Kutta numerical integration aims to minimize these inherent inaccuracies. In particular, the heart has been shown to possess a highly locally isotropic fiber architecture, such that fiber crossing and other bending, branching, etc. is effectively a non-issue (as described by Sosnovik 2009 and others). An array of fiber orientation distribution models exists which fit to more than one peak direction (enabling modeling of crossing fibers, whereas the diffusion tensor model is limited to one peak direction) (described in Garyfallidis 2014 and on the DIPY library website), but these

methods are not particularly relevant to DTMRI in the heart, which does not possess the architectural complexity of, say, the brain.

Comment 9: “Resampling to a 1:4 scale for the purpose of an in vitro model”: how can this scaling affect the biomechanical properties or the fidelity of fiber alignment when translated to a bioprinted construct? Is it possible that changes in scale can alter the geometric and mechanical characteristics? If so, this might impact the interpretation of the results?

For example, when the authors discuss using different gauge needles (22 GA in 1:4; and 18 GA in the 1:1 scale), different actual material diameters are reported on data obtained via micro-CT. Maybe, one could mention that the differences in printed filament thickness are expected to impact the biomechanical properties or alignment fidelity of the final construct, especially for scaling up from a model to a realistic tissue size.

Response:

We thank the reviewer for the question. There are two main points to be considered: biomechanical properties and model fidelity.

Regarding biomechanical properties, the material properties of the silicone ink used for fabrication do not change with scale, the same ink is used for all scales. Please refer to Reviewer #1, Methods Comment #7 for additional comments on the mechanical properties of our models and including additional suggestions in the discussion.

Regarding model fidelity, scaling does not impact streamline geometry at all. Each toolpath fabricated is still directly sourced from the raw DTMRI data and therefore is fully representative of the native fiber architecture of the myocardium. Scaling will impact the amount of detail captured by the model. At smaller scales, fewer streamlines are converted to toolpaths. As we have done, one way to compensate for this is to decrease the diameter of the needle used for printing, which enables more toolpaths to be fabricated. Overall, there is no impact on model fidelity, but there may be a change in the level of anatomical detail captured by the model at different scales.

Comment 10: To eliminate overlapping streamlines, using a bounding cylinder may not be the optimal solution as it does not fully account for complex fiber curvatures (critical when fibers cross or twist). So, how does the algorithm handle situations where the curvature of the fiber significantly deviates from linearity within a local region? Is this going to affect the accuracy of the final toolpath set?

Response:

We appreciate the reviewer's concern regarding the validity of the bounding cylinder method used to reduce the tract set. The bounding cylinder method does in fact account for local nonlinearity in fiber architecture, as the central axis of each cylinder is defined as the corresponding fiber tract itself. As such, the bounding cylinders themselves are curved exactly along the corresponding tract, as shown in main Figures 2c and 2d, and Supplementary Figures 7 and 8. In other words, the cylinders are constructed by “sweeping” a circular cross section along the corresponding tract itself. The exact mathematical definition of the bounding cylinder method is given in Section 2.4.1 (Cylinder/streamline intersection) of the Supplementary Methods.

Comment 11: Can the authors explain if preferring the longest streamline first would generate inaccuracy in replication? Namely, can this bias the structure toward certain spatial regions? Is there any risk to potentially obtain uneven toolpath distribution (example: in areas where fiber lengths naturally vary a lot)?

Response:

We thank the reviewer for the question. While the longest streamline is selected first during the sweep exclusion process, shorter fiber tracts are still existent and eligible for future selection. Longest path first selection increases the average length of the final toolpath set (as the MDF distance criterion for subsequent path selection favors paths of similar length), while at the same time enabling shorter paths to fill spatial gaps that may exist. Choosing a shorter path first does not have a large impact on spatial coverage of the final toolpath set, but does markedly decrease the average toolpath length. In our testing, a smaller average toolpath length led to poorer contiguity and adhesion in the printed model, so longest path first selection was adopted for the NAATIV3 framework. Data supporting this claim has been added to Section 2.4.4 (Initial path selection) of the Supplementary Methods, which is referred to in the main text Methods section.

Comment 12: "structural validity of the set is maintained" after thinning: is this validated in any quantitative metrics or specific methods? Could the authors make this claim more robust by including quantitative measures demonstrating fiber/streamlines patterns preservation? Would it be possible to compare angle distributions before and after thinning, for example?

Response:

We appreciate this question from the reviewer. To substantiate our claim, we have provided additional data in Supplementary Fig. 3. A qualitative depiction is offered in Supplementary Fig. 3A, where we show the distribution of helical angles in the thinned set. The trends present in Fig. 2 A-B are also present in the thinned set. To capture this quantitatively, we plotted a normalized histogram of helical angle before and after the sweep exclusion step (Supplementary Fig. 3B). While there is some decrease in the proportion of streamlines with midmyocardial helical angles around 0°, otherwise the trend is quite consistent across the spectrum of helical angles. Together, we believe this additional data makes a robust case for the validity of the toolpath set being maintained after thinning.

Comment 13: Another idea to substantiate the discussion of these results could be elaborate on the advantages and disadvantages of prioritizing longer streamlines to increase toolpath continuity. This is surely reasonable and intelligent, but could this not necessarily result in always the optimal path for minimizing print time or material use? Is there any perspective on more nuanced approaches for example, balancing length with spatial distribution?

Response:

We thank the reviewer for the comment. As discussed in the response to Comment 11, data has shown that prioritizing longer streamlines does increase average toolpath length while retaining spatial coverage, thus increased contiguity (and therefore structural integrity) in the printed model. While more nuanced approaches, such as balancing length with spatial distribution, could be explored in future work, the current aim of NAATIV3 centers on producing coherent physical models, which is best-served by applying the longest first path method.

Comment 14: Using the inner diameter rather than the outer diameter (OD) should reduce the number of calculated dependencies. Will this not result in underestimating collision events/risks? In complex/ densely packed structures (like myocardial tissue), could such an approximation result in unintended overlaps? Thus leading to structural defects?

Response:

We appreciate the reviewer raising this point, which is one we also considered when designing NAATIV3. The wall thickness of the needles used for printing is a fraction of a millimeter. Accounting for this small width decreased the number of toolpaths in the final set by 64%. Data from preliminary testing with a 22 GA needle are given in the table below.

Both scenarios were tested with 3D printing, and no collisions or damage to the model were noticed when using the inner diameter model. This was further confirmed via micro-CT analysis of our final model, which showed high accuracy to the G-code, suggesting that any interference happening during printing is minor and has little overall impact on the printed structure. In fact, small amounts of touching or overlap between lines of extruded material may be beneficial in holding the printed model together. Because of the substantial benefit from making the zero-width assumption, we decided that adopting the assumption was in the best interest of the model.

Nozzle Diameter Used	Toolpaths in Final Set	Volumetric Coverage of Final Set
Outer diameter (true diameter)	101	1.9%
Inner diameter	280	5.9%

Comment 15: The choice of the cut plane is user-defined, which introduces subjectivity and variability. On this point, the authors can further elaborate in the discussion section. Would it be desirable in the future to find an automated optimization algorithm to determine the optimal plane location, based on fiber orientation analysis rather than manual input?

Response:

We thank the reviewer for this suggestion. During the development of NAATIV3, we explored the option of using quantifiable metrics to guide cut plane placement. However, our data set was not large enough to conclude whether such metrics would be effective or generalizable. It is likely the case that the determination of “optimal” will vary between applications and would still require some degree of customization by the user. We have added language in lines 233-236 to communicate this to the reader and offer the suggestion for future studies of this nature.

Comment 16: It seems that the iterative stochastic optimization for the largest acyclic subgraph is computationally expensive. Would it be desirable to attempt heuristic methods (or graph simplification?) to reduce the computational load. ML could also predict problematic cycles if graph topology is given. Again, this problem could be emphasized in the discussion.

Response:

We appreciate the reviewer’s comment regarding the computationally expensive stochastic method used for acyclic subgraph selection. In developing the NAATIV3 framework, attempts to derive heuristic methods were made. When compared to the stochastic method, it was found that these alternative heuristic methods performed notably worse, such that their increase in computational efficiency was not justified. A description of these alternative methods, and quantitative analysis of their performance has been added to the Supplementary Methods (Section 2.7.1, Alternative heuristic methods and comparison). Perspectives on further development of such methods have been added to the discussion in lines 376-380.

Comment 17: Several toolpaths in the 1:1 model ended up to be removed: may this compromise the accuracy or fidelity of the final printed construct? Is the loss of these paths going to affect the mechanical or functional properties of the model?

Response:

The reviewer correctly states that interference resolution required toolpaths from the 1:1 scale model. We found that the streamlines were particularly removed from the midwall region. This can be seen in the inset in Figure 4E. As a result, the 1:1 model was unstable after removal from the support material used during printing. We have added additional discussion of this, and suggestions for future improvements, in lines 288-291 and 367-375.

Comment 18: "... incur minimal impact on the integrity of the final construct". Can the author point out to the reader which figure or data or reference validate this statement. I suggest the author substantiate this claims with some observations or comments to the related figure.

Response:

We have added an in-text reference to Figure 3D, which shows that relatively few toolpaths are removed during interference resolution. The language has been clarified to reflect that this supports our claim of minimal impact on the final construct.

Comment 19: In the Discussion: "By transcending the planar paradigm and using native fiber tracts to inform toolpath generation, it is possible to 3D bioprint tissue models with a near-native fiber structure for in vitro research, preclinical modeling, and surgical planning." This statements sounds to me as an overclaim, as in this paper no actual 3D bioprinting of tissue model took place, and "near-native fiber structure" as referred to muscle tissue is misleading. "Fiber" is typically the myofiber, which is a structure way below the resolution of 3Dbioprinting (see point 2). I suggest rephrasing this sentence for fair claim and clarity on the actual achievement of this paper. The concept here expressed can also be kept but requires re-elaboration in terms of perspective (e.g., "hold potential for.." or similar phrasing).

Response:

We acknowledge the reviewer's point and have corrected the language as suggested.

Comment 20: Line 309- discussion: "...inability to realistically capture native fiber structure", same comments as above.. I suggest the authors refers to mesoscale architecture of the tissue.

Response:

We appreciate the reviewer's suggestion and have corrected the language as suggested.

Comment 21: The claim that "any bioink could be substituted" lacks experimental validation. The results are obtained with silicone-based ink and so these results may not directly translate to cell-laden bioinks (more prone to shear stress damage and settling in the FRESH bath). If the author will not provide validation data concerning the statements related to "actual bioprinting conditions", then it would be better to elaborate on the current lack of experimental data and open perspectives for future works which could actually prove the technique validity in a real tissue engineering work.

Response:

We thank the reviewer for this suggestion and acknowledge that the lack of cellularization represents an experimental limitation. We have substantially increased the discussion of this limitation and the opportunity for future work in the discussion section (lines 336-347).

In preliminary experimental work in our lab, we have found that the 3D bioprinting process can be adjusted to promote successful cell printing. First, we reduce the needle diameter, which accounts for hydrogel swelling after extrusion and increases the effects of shear alignment. Second, because hydrogel-based bioinks are less viscous than silicone-based inks, we are able to reduce the extrusion pressure and increase the printing speed, simultaneously reducing stress on the cells and reducing total printing time. We have not experienced any issues with cells or hydrogel extrusions settling in the FRESH support. However, we have not quantitatively validated the print fidelity of this left ventricle model with other ink formulations than the silicone used here.

Comment 22: The reported deviation is 0.138 ± 0.250 mm, and the max. dev. is 3.9mm. Is this correct? This significantly exceeds the width of the extruded line (0.54 mm). This max deviation looks to be substantial. Would this compromise the anatomical fidelity of the printed model? If so, how can this be potentially corrected?

Response:

The reviewer has correctly stated the mean, standard deviation, and maximum deviation. The reason for the large maximum deviation is 2 or 3 toolpaths at the apex of the left ventricle model (the top of the model in the printing orientation) that appear to have floated or shifted out of place between printing and imaging with micro-CT. We hypothesize that this occurred while transporting the model between our lab, where it was 3D printed, and the imaging facility. This is discussed in lines 303-306. Due to the fact that the error is highly localized to just a couple of toolpaths, we do not believe that there is any negative impact on the anatomical fidelity of the printed model.

Comment 23: Authors claim successful printability and shape accuracy of the printed model, but I could not find any mechanical or functional tests to evaluate the integrity of the printed structure. If so, then the concern would be that the model's mechanical strength, flexibility, and fiber orientation accuracy are crucial, particularly when aiming to replicate biological tissue properties. Verifying geometric accuracy alone is insufficient for applications such as surgical planning or functional tissue engineering. As for the biological validation (if the goal is to use cell-laden bioinks), mechanical testing (e.g., tensile strength, compressive modulus) to assess whether the printed model behaves similarly to native tissue would be recommended. However, as this could go beyond the scope of the work, it is recommended that the authors use this point to better discuss the limitations of their achievements and provide realistic/pragmatic outlooks on the next steps in this research.

Response:

We appreciate the reviewer's comments. Upon curing and drying the 3D-printed, 1:4-scale model, we were able to handle it without any breakdown of the model. After handling, the model was able to hold water when filled in a simple test. Additional language has been added in lines 281-284 to describe these observations. Please refer to our Response #17 for discussion of the physical properties of the 1:1 scale model.

The question of mechanical behavior and modeling potential are similar to a comment from Reviewer #1. Please refer to Reviewer #1, Methods Comment #7 for additional discussion of the mechanical properties of our models and including more information in the discussion.

Reviewer #1

The authors have addressed most of my concerns and the revised paper, particularly Figure 5, is much improved.

Response:

We are glad to have addressed the reviewer's concerns satisfactorily. Their comments and questions helped us greatly improve our manuscript.

Reviewer #2

I would like to thank the authors for their thoughtful and thorough revisions. I am very pleased with the changes applied to the manuscript, which I believe have significantly improved the clarity, structure, and overall quality of the work. The authors addressed my comments with impressive accuracy and care, demonstrating both scientific rigor and attention to detail. I consider the manuscript ready for publication in its current form.

I appreciate this work, whose major claim is the development and demonstration of the NAATIV3 framework, which enables the translation of DTMRI fiber orientation data into executable toolpaths for 3D bioprinting. The authors unlock the fabrication of tissue-engineered constructs with biologically relevant fiber alignments, directly informed by native tissue architecture. This is a novel and timely contribution to the field of biofabrication, as existing approaches often rely on indirect, heuristic, or simplified representations of anisotropy rather than direct mapping from imaging data. The main value in publishing this article will be providing the readers with a tool that finally bridges the gap between directional imaging and 3D printing. This method not only adds a new computational tool but also opens up opportunities for higher-fidelity modeling of tissue structure-function relationships. The claim is strongly supported by a well-structured algorithmic pipeline.

The work is convincing, and it is likely to influence thinking in both the bioprinting and computational tissue engineering communities. The authors provide compelling evidence of the accuracy and generalizability of their method, and their (now extended) discussion adds to the significance. The research fills a clear gap in current methodologies, and its interdisciplinary nature will likely attract interest beyond the immediate field.

The authors provided an interesting contribution in the field of accelerating biofabrication, where an urgent need is present. The integration of high-resolution imaging data with bioprinting toolpath generation is still in its early stages, and this paper provides a concrete, scalable solution. I believe it will stimulate further innovations at the interface of imaging, modeling, and fabrication, and contribute meaningfully to the development of functionally and structurally realistic engineered tissues.

Response:

We thank the reviewer for their kind words about the work we have shared. We are glad to have addressed the reviewer's concerns with our edits and note that the manuscript has been greatly improved by doing so.